# ON THE DUALITY BETWEEN CONTRASTIVE AND NON-CONTRASTIVE SELF-SUPERVISED LEARNING

**Quentin Garrido**[1,2*]  **Yubei Chen**[1,5]  **Adrien Bardes**[1,3]

**Laurent Najman**[2]  **Yann LeCun**[1,4,5]

[1]Meta AI - FAIR
[2]Univ Gustave Eiffel, CNRS, LIGM, F-77454 Marne-la-Vallée, France
[3]Inria, École normale supérieure, CNRS, PSL Research University
[4]Courant Institute, New York University
[5]Center for Data Science, New York University

## ABSTRACT

Recent approaches in self-supervised learning of image representations can be categorized into different families of methods and, in particular, can be divided into contrastive and non-contrastive approaches. While differences between the two families have been thoroughly discussed to motivate new approaches, we focus more on the theoretical similarities between them. By designing contrastive and covariance based non-contrastive criteria that can be related algebraically and shown to be equivalent under limited assumptions, we show how close those families can be. We further study popular methods and introduce variations of them, allowing us to relate this theoretical result to current practices and show the influence (or lack thereof) of design choices on downstream performance. Motivated by our equivalence result, we investigate the low performance of SimCLR and show how it can match VICReg's with careful hyperparameter tuning, improving significantly over known baselines. We also challenge the popular assumption that non-contrastive methods need large output dimensions. Our theoretical and quantitative results suggest that the numerical gaps between contrastive and non-contrastive methods in certain regimes can be closed given better network design choices and hyperparameter tuning. The evidence shows that unifying different SOTA methods is an important direction to build a better understanding of self-supervised learning.

## 1 INTRODUCTION

Self-supervised learning (SSL) of image representations has shown significant progress in the last few years (Chen et al., 2020a; He et al., 2020; Chen et al., 2020b; Grill et al., 2020; Lee et al., 2021b; Caron et al., 2020; Zbontar et al., 2021; Bardes et al., 2021; Tomasev et al., 2022; Caron et al., 2021; Chen et al., 2021b; Li et al., 2022a; Zhou et al., 2022a;b; HaoChen et al., 2021), approaching, and sometime even surpassing, the performance of supervised baselines on many downstream tasks. Most recent approaches are based on the joint-embedding framework with a siamese network architecture (Bromley et al., 1994) which are divided into two main categories, contrastive and non-contrastive methods. Contrastive methods bring together embeddings of different views of the same image while pushing away the embeddings from different images. Non-contrastive methods also attract embeddings of views from the same image but remove the need for explicit negative pairs, either by architectural design (Grill et al., 2020; Chen & He, 2020) or by regularization of the covariance of the embeddings (Zbontar et al., 2021; Bardes et al., 2021; Li et al., 2022b).

While contrastive and non-contrastive approaches seem very different and have been described as such (Zbontar et al., 2021; Bardes et al., 2021; Ermolov et al., 2021; Grill et al., 2020), we pro-

---

*Correspondence to `garridoq@meta.com`

pose to take a closer look at the similarities between the two, both from a theoretical and empirical point of view and show that there exists a close relationship between them. We focus on covariance regularization-based non-contrastive methods (Zbontar et al., 2021; Ermolov et al., 2021; Bardes et al., 2021) and demonstrate that these methods can be seen as contrastive between the dimensions of the embeddings instead of contrastive between the samples. We, therefore, introduce the term *dimension-contrastive* methods which we believe is better suited for them, and refer to the original contrastive methods as *sample-contrastive* methods. To show the similarities between the two, we define contrastive and non-contrastive criteria based on the Frobenius norm of the Gram and covariance matrices of the embeddings, respectively, and show the equivalence between the two under assumptions on the normalization of the embeddings. We then relate popular methods to these criteria, highlighting the links between them and further motivating the use of the *sample-contrastive* and *dimension-contrastive* nomenclature. Finally, we introduce variations of an existing dimension-contrastive method (VICReg), and a sample-contrastive one (SimCLR). This allows us to verify this equivalence empirically and improve both VICReg and SimCLR through this lens.

Our contributions can be summarized as follows:

- We make a significant effort to unify several SOTA sample-contrastive and dimension-contrastive methods and show that empirical performance gaps can be closed completely. By pinpointing its source, we consolidate our understanding of SSL methods.

- We introduce two criteria that serve as representatives for sample- and dimension-contrastive methods. We show that they are equivalent for doubly normalized embeddings, and then relate popular methods to them, highlighting their theoretical similarities.

- We introduce methods that interpolate between VICReg and SimCLR to study the practical impact of precise components of their loss functions. This allows us to validate empirically our theoretical result by isolating the sample- and dimension-contrastive nature of methods.

- Motivated by the equivalence, we show that advantages attributed to one family can be transferred to the other. We improve SimCLR's performance to match VICReg's, and improve VICReg to make it as robust to embedding dimension as SimCLR.

## 2 RELATED WORK

**Sample-contrastive methods.** In self-supervised learning of image representations, contrastive methods pull together embeddings of distorted views of a single image while pushing away embeddings coming from different images. Many works in this direction have recently flourished (Chen et al., 2020a; He et al., 2020; Chen et al., 2020b; 2021b; Yeh et al., 2021), most of them using the InfoNCE criterion (Oord et al., 2018), except HaoChen et al. (2021), that uses squared similarities between the samples. Clustering-based methods (Caron et al., 2018; 2020; 2021) can be seen as contrastive between prototypes, or clusters, instead of samples.

**Non-contrastive methods.** Recently, methods that deviate from contrastive learning have emerged and eliminate the use of negative samples in different ways. Distillation-based methods such as BYOL (Grill et al., 2020), SimSiam (Chen & He, 2020) or DINO (Caron et al., 2021) use architectural tricks inspired by distillation to avoid the collapse problem. Information maximization methods (Bardes et al., 2021; Zbontar et al., 2021; Ermolov et al., 2021; Li et al., 2022b) maximize the informational content of the representations and have also had significant success. They rely on regularizing the empirical covariance matrix of the embeddings so that their informational content is maximized. Our study of dimension-contrastive learning focuses on these covariance-based methods.

**Understanding contrastive and non-contrastive learning.** Recent works tackle the task of understanding and characterizing methods. The fact that a method like SimSiam does not collapse is studied in Tian et al. (2021). The loss landscape of SimSiam is also compared to SimCLR's in Pokle et al. (2022), which shows that it learns bad minima. In Wang & Isola (2020), the optimal solutions of the InfoNCE criterion are characterized, giving a better understanding of the embedding distributions. A spectral graph point of view is taken in HaoChen et al. (2022; 2021); Shen et al. (2022) to analyze self-supervised learning methods. Practical properties of contrastive methods have been studied in Chen et al. (2021a). In Huang et al. (2021) Barlow twins criterion is shown to be related to an upper bound of a sample-contrastive criterion. We go further and exactly quantify the gap between the criterion, which allows us to use the link between methods in practical scenarios. Barlow

Twins' criterion is also linked to HSIC in Tsai et al. (2021). The use of data augmentation in sample-contrastive learning has also been studied from a theoretical standpoint in Huang et al. (2021); Wen & Li (2021). In Balestriero & LeCun (2022), popular self-supervised methods are linked to spectral methods, providing a unifying framework that highlights their differences. The gradient of various methods is also studied in Tao et al. (2021), where they show links and differences between them. In Lee et al. (2021a), a link is made between CCA and SCL by showing similar error bounds on linear classifiers.

## 3 EQUIVALENCE OF THE CONTRASTIVE AND NON-CONTRASTIVE CRITERION

While our results only depend on the embeddings and not the architecture used to obtain them, nor do they depend on the data modality, all the studied methods are placed in a joint embedding framework and applied on images. Given a dataset $\mathcal{D}$ with individual datum $d_i \in \mathbb{R}^{c \times h \times w}$, this datum is augmented to obtain two views $x_i$ and $x_i'$. These two views are then each fed through a pair of neural networks $f_\theta$ and $f_{\theta'}'$. We obtain the *representations* $f_\theta(x_i)$ and $f_{\theta'}'(x_i')$, which are fed through a pair of projectors $p_\theta$ and $p_{\theta'}'$ such that *embeddings* are defined as $p_\theta(f_\theta(x_i))$ and $p_{\theta'}'(f_{\theta'}'(x_i'))$. We denote the matrices of embeddings $\mathcal{K}$ and $\mathcal{K}'$ such that $\mathcal{K}_{\cdot,i} = p_\theta(f_\theta(x_i))$, and similarly for $\mathcal{K}'$, we have $\mathcal{K} \in \mathbb{R}^{M \times N}$, with $M$ the embedding size and $N$ the batch size, and similarly for $\mathcal{K}'$. These embedding matrices are the primary object of our study. In practice, we use $f_\theta = f_{\theta'}'$ and $p_\theta = p_{\theta'}'$. While most self-supervised learning approaches use positive pairs $(x_i, x_i')$ and negative pairs $\{\forall j, j \neq i, (x_i, x_j)\} \bigcup \{\forall j, j \neq i, (x_i, x_j')\}$ for a given view $x_i$, we focus on the simpler scenario where negative samples are just $\{\forall j, j \neq i, (x_i, x_j)\}$. There is no fundamental difference when $\theta = \theta'$ and when the same distribution of augmentations is used for both branches, and we therefore make these simplifications to make the analysis less convoluted.

We start by defining precisely which contrastive and non-contrastive criteria we will be studying throughout this work. These criteria will be used to classify methods in two classes, *sample-contrastive*, which corresponds to what is traditionally thought of as contrastive, and *dimension-contrastive*, which will encompass non-contrastive methods relying on regularizing the covariance matrix of embeddings.

**Definition 3.1.** Given a matrix $A \in \mathbb{R}^{n \times n}$. We define its *extracted diagonal* diag $(A) \in \mathbb{R}^{n \times n}$ as:

$$\text{diag}\,(A)_{i,j} = \begin{cases} A_{i,i}, & \text{if } i = j \\ 0, & \text{otherwise.} \end{cases} \tag{1}$$

**Definition 3.2.** A method is said to be *sample-contrastive* if it minimizes the contrastive criterion $L_c = \|\mathcal{K}^T\mathcal{K} - \text{diag}(\mathcal{K}^T\mathcal{K})\|_F^2$. Similarly, a method is said to be *dimension-contrastive* if it minimizes the non-contrastive criterion $L_{nc} = \|\mathcal{K}\mathcal{K}^T - \text{diag}(\mathcal{K}\mathcal{K}^T)\|_F^2$.

The *sample-contrastive* criterion can be seen as penalizing the similarity between different pairs of images, whereas the *dimension-contrastive* criterion can be seen as penalizing the off-diagonal terms of the covariance matrix of the embeddings. These criteria respectively try to make pairs of samples or dimensions orthogonal.

**Invariance criterion.** While $L_c$ and $L_{nc}$ focus on regularizing the embedding space, they are not optimized alone. They are usually combined with an invariance criterion that aims at producing the same embedding for two views of the same image. As such, a complete self-supervised loss would look like $L_{SSL} = L_{inv} + L_{reg}$ with $L_{reg}$ being either $L_c$ or $L_{nc}$ for our prototypical sample-contrastive and dimension-contrastive methods. This invariance criterion is generally a similarity measure, such as the cosine similarity or the mean squared error of the difference between a positive pair of samples. Both are equivalent from an optimization point of view if using normalized embeddings, hence we focus on the regularization part which is the source of differences between these methods.

**Proposition 3.1.** Considering an infinite amount of available negative samples, SimCLR and DCL's criteria lead to embeddings where for negative pairs $(x, x^-) \in \mathbb{R}^M$ we have

$$\mathbb{E}\left[x^T x^-\right] = 0 \quad \text{and} \quad \text{Var}\left[x^T x^-\right] = \frac{1}{M}. \tag{2}$$

SimCLR and DCL cannot be easily linked to $L_c$ since they rely on cosine similarities instead of their square or absolute value. Indeed, while $L_c$ aims at making pairs of embeddings or dimensions

orthogonal, SimCLR and DCL's criteria go a step further and aim at making them opposite. Both cannot be satisfied perfectly in practice, as we would need as many dimensions as samples for $L_c$ to have all negative pairs be orthogonal, and more than two vectors cannot be pairwise opposite for SimCLR and DCL's criterion. Nonetheless, as shown by Proposition 3.1, SimCLR and DCL's criteria will lead to dot products of negative pairs with a null mean, which is exactly the aim of $L_c$. This shows that while the original formulations of DCL and SimCLR do not fit perfectly into our theoretical framework, they will still lead to results similar to other methods that we study. In order to complement this result, we introduce SimCLR-sq and SimCLR-abs as variations of SimCLR, which respectively use square or absolute values of cosine similarities. We define DCL-sq and DCL-abs similarly. We provide a study of SimCLR-sq and SimCLR-abs in supplementary section G, where we compare them to SimCLR. The main conclusion is that the distribution of off-diagonal terms of the Gram matrix is similar between all studied methods, with a high concentration of values around zero, as predicted by Proposition 3.1. We also see that changing SimCLR into these variations does not impact performance. We even see a small increase in top-1 accuracy on ImageNet (Deng et al., 2009) with linear evaluation when using SimCLR-abs, where we reach 68.71% top-1 accuracy, compared to 68.61% with our improved reproduction of SimCLR. Both of these theoretical and practical arguments reinforce the proximity of SimCLR to our framework.

**Proposition 3.2.** SimCLR-abs/sq, DCL-sq/abs, and Spectral Contrastive Loss (HaoChen et al., 2021) are sample-contrastive methods. Barlow Twins (Zbontar et al., 2021), VICReg (Bardes et al., 2021) and TCR (Li et al., 2022b) are dimension-contrastive methods.

Even though they do not fit perfectly in our framework, we discuss how methods such as DINO, SimSiam, or MoCo can be linked to $L_c$ and $L_{nc}$ in supplementary section B. From proposition 3.2 we can see that sample-contrastive and dimension-contrastive methods can respectively be linked together by $L_c$ and $L_{nc}$. This alone is not enough to show the link between those two families of methods and we will now discuss the link between $L_c$ and $L_{nc}$ to show how close those families are.

**Theorem 3.3.** The sample-contrastive and dimension-contrastive criteria $L_c$ and $L_{nc}$ are equivalent up to row and column normalization of the embedding matrix $\mathcal{K}$. Consider a batch size of $N$ and an embedding dimension of $M$. We have:

$$L_{nc} + \sum_{j=1}^{M} \|\mathcal{K}_{j,\cdot}\|_2^4 = L_c + \sum_{i=1}^{N} \|\mathcal{K}_{\cdot,i}\|_2^4. \qquad (3)$$

Theorem 3.3 is similar to lemma 3.2 from Le et al. (2011), where we consider matrices that are not doubly stochastic. It is worth noting that our result does not rely on any assumption about the embeddings themselves. A similar result was also used recently in HaoChen et al. (2022), where they relate the spectral contrastive loss to $L_{nc}$.

The proof of theorem 3.3 hinges on the fact that the squared Frobenius norm of the Gram and Covariance matrix of the embeddings are equal, i.e., $\|\mathcal{K}^T\mathcal{K}\|_F^2 = \|\mathcal{K}\mathcal{K}^T\|_F^2$. This means that penalizing all the terms of the Gram matrix (i.e., pairwise similarities) is the same as penalizing all of the terms of the Covariance matrix. While this gives an intuition for the similarity between the contrastive and non-contrastive criteria, it is not as representative of the criteria used in practice as $L_c$ and $L_{nc}$ are. While theorem 3.3 shows that sample-contrastive and dimension-contrastive approaches minimize similar criteria, for none of these methods can we conclude that both criteria can be used interchangeably. However, if both rows and columns of $\mathcal{K}$ were L2 normalized, we would have $L_{nc} = L_c + N - M$. In this case, both criteria would be equivalent from an optimization point of view, and we could conclude that sample-contrastive and dimension-contrastive methods are all minimizing the same criterion.

**Influence of normalization.** The difference between the two criteria then lies in the embedding matrix row and column norms, and most approaches do normalize it in one direction. Since SimCLR relies on the cosine distance as a similarity measure between embeddings, we can effectively say that it uses normalized embeddings. Similarly, Spectral Contrastive Loss projects the embeddings on a ball of radius $\sqrt{\mu}$, with $\mu$ a tuned parameter, meaning that the embeddings are normalized before the computation of the loss function.

Barlow Twins normalizes dimensions such that they have a null mean and unit variance, so all dimensions will have a norm of $\sqrt{N}$. VICReg takes a similar approach where dimensions are centered, but their variance is regularized by the variance criterion. This is similar to what is done for Barlow Twins and thus leads to dimensions with constant norm. However, for TCR, the embeddings

are normalized and not the dimensions, contrasting with other dimension-contrastive methods.
One of the main differences between normalizing embeddings or dimensions is that in the former case, embeddings are projected on a $M - 1$ dimensional hypersphere, and in the latter, they are not constrained on a particular manifold; instead, their spread in the ambient space is limited.

Nonetheless, a constraint on the norm of the embeddings also constrains the norm of the dimensions indirectly, and vice versa, as illustrated in lemma 3.4.

**Lemma 3.4.** If embeddings are normalized such that $\forall i, \|\mathcal{K}_{\cdot,i}\|_2 = a$ we have

$$\frac{N^2}{M} a^4 \leq \sum_{j=1}^{M} \|\mathcal{K}_{j,\cdot}\|_2^4 \leq N^2 a^4. \tag{4}$$

Conversely, if dimensions are normalized such that $\forall j, \|\mathcal{K}_{j,\cdot}\|_2 = a$ we have

$$\frac{M^2}{N} a^4 \leq \sum_{i=1}^{N} \|\mathcal{K}_{\cdot,i}\|_2^4 \leq M^2 a^4. \tag{5}$$

Following the proof of lemma 3.4, the lower bounds can be constructed with a constant embedding matrix and the upper bounds with an embedding matrix where either the rows or columns contain only one non-zero element. Both correspond to collapsed representations and will thus not be attained in practice. While it is impossible to characterize non-collapsed embedding matrices and, as such, derive better practical bounds, these bounds can still be useful to derive the following corollary. We study how close methods are to these bounds in practice in section H of the supplementary material. The main conclusion is that in all practical scenarios, the sum of norms will be very close to the lower bounds, deviating by a single-digit factor.

**Corollary 3.4.1.** If embeddings are L2-normalized we have

$$L_{nc} - N + \frac{N^2}{M} \leq L_c \leq L_{nc} - N + N^2. \tag{6}$$

Similarly, if dimensions are L2-normalized we have

$$L_c - M + \frac{M^2}{N} \leq L_{nc} \leq L_c - M + M^2. \tag{7}$$

Lemma 3.4 applied to Theorem 3.3 directly gives us corollary 3.4.1, which means that in practical scenarios, even when we compare methods where the embeddings are not doubly normalized, the contrastive and non-contrastive criteria can't be arbitrarily far apart. We further show experimentally in section 5.1 that the normalization strategy does not matter from a performance point of view on SimCLR, reinforcing this argument. Considering the previous discussions, we thus argue that the main differences between sample-contrastive and dimension-contrastive methods come from the optimization process as well as the implementation details.

**Disguising VICReg as a contrastive method.** To illustrate theorem 3.3 we can rewrite VICReg's criterion to make $L_c$ appear. We first recall the different components of VICReg's criterion. The variance criterion $v$ is a hinge loss that aims at making the variance along every dimension greater than 1, and the covariance criterion $c$ is exactly defined as $L_{nc}$ applied to centered embeddings. For more details, confer Bardes et al. (2021). To make $L_c$ appear, we will still apply the invariance and variance criterion on the embeddings, but the covariance criterion will be applied to the transposed embeddings, effectively making it contrastive since we have:

$$c(\mathcal{K}^T) = \|\mathcal{K}^T \left(\mathcal{K}^T\right)^T - \mathrm{diag}\left(\mathcal{K}^T \left(\mathcal{K}^T\right)^T\right)\|_F^2 = \|\mathcal{K}^T \mathcal{K} - \mathrm{diag}(\mathcal{K}^T \mathcal{K})\|_F^2 = L_c(\mathcal{K}). \tag{8}$$

We then just need to add a regularization term on the norms of embeddings and dimensions as follows:

$$L_{reg}(\mathcal{K}) = \sum_{i=1}^{N} \|\mathcal{K}_{\cdot,i}\|_2^4 - \sum_{j=1}^{M} \|\mathcal{K}_{j,\cdot}\|_2^4, \tag{9}$$

and VICReg's loss function can then be written as

$$\mathcal{L}_{VICReg} = \lambda \sum_{i=1}^{N} \|\mathcal{K}_{\cdot,i} - \mathcal{K}'_{\cdot,i}\|_2^2 + \mu \left(v(\mathcal{K}) + v(\mathcal{K}')\right) + \nu \left(L_c(\mathcal{K}) + L_{reg}(\mathcal{K}) + L_c(\mathcal{K}') + L_{reg}(\mathcal{K}')\right). \tag{10}$$

This rewriting can be seen as a variation of SCL to which is added $L_{reg}$ and that uses the variance loss for normalization. Being able to make VICReg's criterion sample-contrastive highlights the close relationship between existing sample-contrastive and dimension-contrastive methods and further shows that the difference in the behavior of different methods is not mainly due to whether they are contrastive or not.

## 4 INTERPOLATING BETWEEN METHODS: IMPACT OF THE LOSS FUNCTION.

While we have discussed the link between the contrastive and non-contrastive criteria, we can wonder how the design differences in popular criteria manifest themselves in practice. To do so we start by introducing variations on VICReg that will allow us to interpolate between VICReg and SimCLR while isolating precise components of the loss function. While our focus will be on performance, we provide an analysis of the optimization quality in supplementary section J. The conclusion is that while some design choices negatively impact the optimization process on the embeddings, there are no easily visible differences in the representations which are used in practice.

**VICReg variations.** We introduce two variants of VICReg, one that is non-contrastive but inspired by the InfoNCE criterion and one that is contrastive and also inspired by the InfoNCE criterion. The former is motivated by one of the main differences between methods, which is the use of the LogSumExp (LSE) for the repulsive force (e.g., SimCLR) or the use of the sum of squares (e.g., SCL, VICreg, BT). The latter is motivated by the wish to design contrastive methods, where implementation details such as the negative pair sampling are as close as possible to another method. This way, comparing VICReg to either of those methods will yield a comparison that truly isolates specific components of the loss function. These two methods can also be seen as a transformation from VICReg to SimCLR, which allows us to see when the behavior of VICReg becomes akin to SimCLR's, as illustrated in the following diagram:

$$\text{VICReg} \xrightarrow{\text{LogSumExp}} \text{VICReg-exp} \xrightarrow{\text{Contrastive}} \text{VICReg-ctr} \xrightarrow{\text{Neg. pair sampling}} \text{SimCLR}$$

The first variant that we will introduce is VICReg-exp, which uses a repulsive force inspired by the InfoNCE criterion. We first define the exponential covariance regularization as:

$$c_{exp}(\mathcal{K}) = \frac{1}{d} \sum_i \log \left( \sum_{j \neq i} e^{C(\mathcal{K})_{i,j}/\tau} \right), \tag{11}$$

VICReg-exp is then VICReg where we replace the covariance criterion by this exponential covariance criterion, giving an overall criterion of

$$\mathcal{L}_{VICReg-exp} = \lambda \sum_{i=1}^{N} \|\mathcal{K}_{\cdot,i} - \mathcal{K}'_{\cdot,i}\|_2^2 + \mu \left( v(\mathcal{K}) + v(\mathcal{K}') \right) + \nu \left( c_{exp}(\mathcal{K}) + c_{exp}(\mathcal{K}') \right). \tag{12}$$

We then define VICReg-ctr, which is VICReg-exp where we transpose the embedding matrix before applying the variance and covariance regularization. This means that the variance regularization will regularize the norm of the embeddings, and the covariance criterion now penalizes the Gram matrix, with the same repulsive force as in DCL. Transposing the embedding matrix for the variance criterion leads to more stable training and enables the use of mixed precision. We thus have the following criterion:

$$\mathcal{L}_{VICReg-ctr} = \lambda \sum_{i=1}^{N} \|\mathcal{K}_{\cdot,i} - \mathcal{K}'_{\cdot,i}\|_2^2 + \mu \left( v(\mathcal{K}^T) + v(\mathcal{K}'^T) \right) + \nu \left( c_{exp}(\mathcal{K}^T) + c_{exp}(\mathcal{K}'^T) \right). \tag{13}$$

This way, VICReg-exp will allow us to study the influence of the use of the LogSumExp operator in the repulsive force, and VICReg-ctr to study the difference between sample-contrastive and dimension-contrastive methods when comparing it to VICReg-exp. We will now be able to study the optimization of the two criteria and see how different design choices affect it.

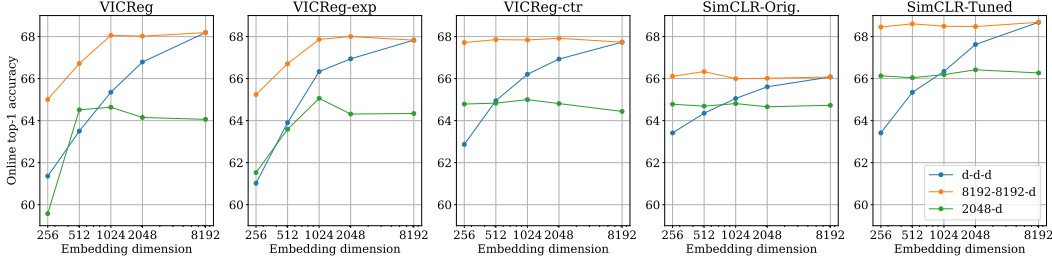

Figure 1: VICReg, VICReg-exp, and VICReg-ctr perform similarly in 100 epochs training, validating empirically our theoretical result. While the original implementation of SimCLR performs significantly worse – which is unexpected per our theory – we are able to improve its performance to VICReg's level. This further validates our findings. While different projector architectures impact performance, behaviors are similar across methods. Confer supplementary section K for numerical values and hyperparameters.

## 5 PRACTICAL DIFFERENCES BETWEEN SAMPLE-CONTRASTIVE AND DIMENSION-CONTRASTIVE METHODS

While we have discussed how close sample and dimension contrastive methods are in theory, one of the primary considerations when choosing or designing a method is the performance on downstream tasks. Linear classification on ImageNet has been the main focus in most SSL methods, so we will focus on this task. We will consider the two following aspects, which are responsible for most of the discrepancies between methods.

**Loss implementation.** Thanks to VICReg-exp, we are able to study the difference between penalizing the Frobenius norm directly and using a LogSumExp to penalize it. Similarly, for VICReg-ctr we are able to study the practical differences between the contrastive and non-contrastive criteria. Finally, with SimCLR we will be able to see how the last details between VICReg-ctr and it can impact performance.

**Projector architecture.** One of the main differences in methods is how the projector is designed. To describe projector architectures we use the following notation: $X - Y - Z$ means that we use linear layers of dimensions $X$, then $Y$ and $Z$. Each layer is followed by a ReLU activation and a batch normalization layer. The last layer has no activation, batch normalization, or bias.
In order to study the impact that this has on performance with respect to embedding size, we study three scenarios. First, $d - d - d$, which is the scenario used for VICReg and BT, then $2048 - d$ which was originally used for SimCLR, and finally $8192 - 8192 - d$ which was optimal for large embeddings with VICReg.

Due to the extensive nature of the following experiments, we use a proxy of the classical linear evaluation on ImageNet, where the classifier is trained alongside the backbone and projector. Representations are fed to a linear classifier while keeping the gradient of this classifier's criterion from flowing back through the backbone. The addition of this linear classifier is extremely cheap and avoids a costly linear evaluation after training. The performance of this online classifier correlates almost perfectly with its offline counterpart, so we can rely on it to discuss the general behaviors of various methods. This evaluation was briefly mentioned in Chen et al. (2020a) but without experimental support. We discuss the correlation between the two further in supplementary section E.

**Empirical validation.** The first takeaway from figure 1 is that the transition VICReg → VICReg-exp via the addition of the LogSumExp did not alter overall performance or behavior. While small performance differences are visible between the two when using light projectors, especially at low embedding dimension, as soon as we use a larger projector these differences disappear with them achieving 68.13% and 68.00% respectively. Focusing on the transition VICReg-exp → VICReg-ctr, we can see that there is no noticeable gap in performance in a setting where we were able to isolate the sample-contrastive and dimension-contrastive nature of the methods. This validates empirically our theoretical findings on the equivalence of sample-contrastive and dimension-contrastive methods. When comparing VICReg-ctr to our reproduction of SimCLR,

Table 1: Normalisation strategy used by different methods. Scenarios A and B for SimCLR enable a fairer comparison to VICReg-ctr and VICReg respectively.

| Method | VICReg | VICReg-exp | VICReg-ctr | SimCLR | | |
| --- | --- | --- | --- | --- | --- | --- |
| | | | | Classical | A | B |
| Dimension centering | ✓ | ✓ | ✓ | ✗ | ✓ | ✓ |
| Embedding norm | | | 1 | 1 | 1 | |
| Dimension norm | $\sqrt{N}$ | $\sqrt{N}$ | | | | $\sqrt{N/M}$ |

using the original hyperparameters, we can see that VICReg-ctr performs significantly better than SimCLR, achieving $67.92\%$ top-1 accuracy compared to $66.33\%$. This is surprising since the main difference between the two is that VICReg-ctr uses fewer negative pairs, which should not improve performance. As such we will focus on showing that the previously known performance of SimCLR is suboptimal and then fix it. In supplementary section F we further validate our results with $k$-nn classification accuracy and also show that features correlate extremely well between methods.

**Improving SimCLR's performance.** To the best of our knowledge, the highest top-1 accuracies reported on ImageNet with SimCLR in 100 epochs are around $66.8\%$ (Chen et al., 2021a). While much higher than the $64.7\%$ originally reported, this is still significantly lower than VICReg. Motivated by the performance of VICReg-ctr, we used the same projector as VICReg and heavily tuned hyperparameters, allowing us to find that a temperature of $0.15$ and base learning rate of $0.5$ can lead to a top-1 accuracy of $68.6\%$, matching VICReg's performance in Bardes et al. (2021). This reinforces our theoretical insights and highlights the contribution of precise engineering[1] in recent self-supervised advances. As it stands, SimCLR can still serve as a strong baseline.

**A larger projector increases performance.** From figure 1 we can see that for every studied method, going from a projector with architecture $2048 - d$ to $8192 - 8192 - d$ yielded a significant boost in performance, especially for VICReg and VICReg-ctr, both gaining $3.5 - 4$ points. The projector $d - d - d$ is in between the two depending on the embedding dimension but also shows a similar trend, the performance increases with the number of parameters for every method. While out of the scope of this work, the study of the importance of the projector's capacity is an exciting line of work that should help gain a deeper understanding of its role in self-supervised learning. We provide a preliminary discussion in the supplementary section I.

**Clearing up misconceptions.** While contrastive methods are often thought of as sample inefficient, thus requiring large batch sizes, and non-contrastive methods as dimension inefficient, thus projectors with large output dimensions, we argue that both of these assumptions are misleading and that all of these apparent issues can be alleviated with some care. Most notably, the need for large batch sizes of contrastive methods has been studied in Yeh et al. (2021) and Zhang et al. (2022) where the main conclusions are that with more tuning of the InfoNCE parameters, the robustness of SimCLR and MoCo to small batches can be improved. Regarding the robustness of non-contrastive methods to embedding dimension, our experiments show that with a more adequate projector architecture and with careful hyperparameter tuning, the drop in performance at low embedding dimension is not as present as initially reported (Zbontar et al., 2021; Bardes et al., 2021). With 256-dimensional embeddings, we were able to achieve $61.36\%$ top-1 accuracy by tuning VICReg's hyperparameters, compared to the $55.9\%$ that were initially reported in Bardes et al. (2021). This can be further improved to $65.01\%$ with a bigger projector. While a drop is still present, we are able to reach peak performance at 1024 dimensions, which is lower than the representation's dimension of 2048 and shows that a large embedding dimension is not a deciding factor in downstream performance.

## 5.1 INFLUENCE OF THE NORMALIZATION STRATEGY

While we have shown that the performance gap between sample-contrastive and dimension-contrastive methods can be closed with careful hyperparameter tuning, in the studied settings not

---

[1]Popular PyTorch implementations of SimCLR that are compatible with DDP use a wrong `gather` operator, which when combined with DDP divides the gradients by the world size. The implementation in VICReg's codebase is correct and should be used. This change had a significant impact on performance and allowed us to reach VICReg's performance.

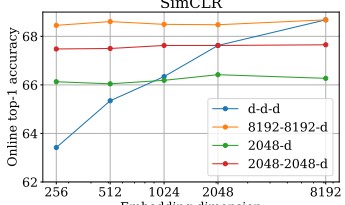 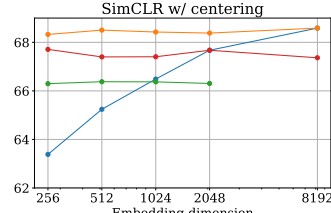 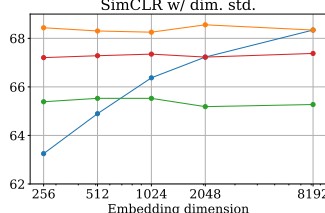

Figure 2: The performance of SimCLR is unchanged when introducing centering or dimension standardization, highlighting the lack of importance of normalization on peak performance.

all details are equal. This is especially true regarding the normalization strategies that are used, and we illustrate the different ones in table 1. In order to show that these differences do not impact performance, we will introduce two variations of SimCLR. First, we will look at SimCLR with the centering of the dimensions, and then at SimCLR with the centering of the dimensions as well as a normalization along the dimensions instead of the embeddings. This last strategy is in essence a standardization of the dimensions and is the same scheme used by VICReg. More precisely the dimension standardization can be written as :

$$\forall i \in [1, \ldots, M] \quad \mathcal{K}_{\cdot,i} = \frac{\hat{\mathcal{K}}_{\cdot,i}}{\|\hat{\mathcal{K}}_{\cdot,i}\|_2} \times \sqrt{\frac{N}{M}} \quad \text{with} \quad \hat{\mathcal{K}}_{\cdot,i} = \mathcal{K}_{\cdot,i} - \frac{1}{N} \sum_{j=1}^{N} \mathcal{K}_{j,i}. \quad (14)$$

These variations will allow us to compare VICReg and SimCLR when both adopt the same normalization strategy, resulting in a comparison that will more closely fit our theoretical framework.

As we can see in figure 2, the centering and dimension standardization do not impact performance at all and we are able to achieve the same peak performance as before. The performance is slightly lower with a shallow projector $2048 - d$, but in all the other scenarios we retrieve the same performance as the original SimCLR. This performance is on par with VICReg and its variations which reinforces our theoretical result in practice. This was further confirmed in a $1000$ epoch run, where SimCLR with dimension standardization was able to reach $72.6\%$ top-1 accuracy, compared to $73.3\%$ for VICReg. While a small difference persists, hyperparameter tuning is very expensive in this setting and is most likely the cause of this gap.

From these results, we can conclude that while the normalization strategy can be theoretically motivated or can ease the optimization process, it is not a deciding factor in the performance of self-supervised methods and that the normalization strategy that should be used is the one that is the easiest to work with for a given method.

## 6 CONCLUSION

Through an analysis of their criteria, we were able to show that sample-contrastive and dimension-contrastive methods have learning objectives that are closely related, as they are effectively minimizing criteria that are equivalent up to row and column normalization of the embedding matrix. This suggests a certain duality in the behavior of such methods, which we studied empirically. Through the lens of variations of VICReg, we were able to study popular design choices in self-supervised loss functions and show their lack of impact on performance, significantly improving the robustness to embedding dimension of VICReg along the way. Motivated by our theoretical findings, we performed ample hyperparameter tuning on SimCLR and were able to close its performance gap with VICReg. We also showed that the normalization strategy does not play an important role in performance. This further reinforces the similarities between methods as predicted by our theoretical results. We expect that our results will help extend theoretical works in self-supervised learning to a wider family of methods, as well as help analyses by deriving criteria that are easier to work with. We also expect that our findings will help alleviate preconceived ideas on contrastive and non-contrastive learning. If one thing must be remembered from this work, it is that *dimension-contrastive and sample-contrastive methods are two sides of the same coin*. Finally, perhaps the most important message of this work is to show that different SOTA SSL methods can be unified. Pinpointing the source of the advancements is an important direction to consolidate our understanding.

## 7 ACKNOWLEDGMENTS

The authors wish to thank Randall Balestrierio, Li Jing, Grégoire Mialon, Nicolas Ballas, Surya Ganguli, and Pascal Vincent, in no particular order, for insightful discussions. We also thank Florian Bordes for the efficient implementations that were used for our experiments.

## 8 REPRODUCIBILITY STATEMENT

While our pretrainings are very costly, each taking around a day with $8$ V100 GPUs, we provide complete hyperparameter values in table S6. They are compatible with official implementations of the losses, and for VICReg-ctr and VICReg-exp we also provide PyTorch pseudocode in supplementary section L. In order to reproduce our main figure, we also give the numerical performance in table S5. All of this should make our results reproducible, and, more importantly, should make it so that practitioners can benefit from the improved performance that we introduce.

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

## A  BACKGROUND

In this section, we will recall the loss functions of all the methods we are considering throughout our theoretical analysis.

**DCL:** We first take a look at DCL's criterion. We consider that $\mathcal{K}$ is l2 normalized column-wise, i.e. embeddings are normalized. We have

$$\mathcal{L}_{\text{DCL}} = \sum_{i=1}^{N} -\log\left(\frac{e^{\mathcal{K}_{\cdot,i}^T \mathcal{K}'_{\cdot,i}/\tau}}{\sum_{j\neq i} e^{\mathcal{K}_{\cdot,i}^T \mathcal{K}_{\cdot,j}/\tau}}\right) = \sum_{i=1}^{N} -\frac{\mathcal{K}_{\cdot,i}^T \mathcal{K}'_{\cdot,i}}{\tau} + \log\left(\sum_{j\neq i} e^{\mathcal{K}_{\cdot,i}^T \mathcal{K}_{\cdot,j}/\tau}\right). \quad (15)$$

**SimCLR:** We now take a look at SimCLR's criterion. We consider that $\mathcal{K}$ is l2 normalized column-wise, i.e. embeddings are normalized. We have

$$\mathcal{L}_{\text{SimCLR}} = \sum_{i=1}^{N} -\log\left(\frac{e^{\mathcal{K}_{\cdot,i}^T \mathcal{K}'_{\cdot,i}/\tau}}{e^{\mathcal{K}_{\cdot,i}^T \mathcal{K}'_{\cdot,i}/\tau} + \sum_{j\neq i} e^{\mathcal{K}_{\cdot,i}^T \mathcal{K}_{\cdot,j}/\tau}}\right) \quad (16)$$

$$= \sum_{i=1}^{N} -\frac{\mathcal{K}_{\cdot,i}^T \mathcal{K}'_{\cdot,i}}{\tau} + \log\left(e^{\mathcal{K}_{\cdot,i}^T \mathcal{K}'_{\cdot,i}/\tau} + \sum_{j\neq i} e^{\mathcal{K}_{\cdot,i}^T \mathcal{K}_{\cdot,j}/\tau}\right). \quad (17)$$

**Spectral Constrastive Loss:** Spectral Contastive Loss is defined as

$$\mathcal{L}_{SCL} = -2\sum_{i=1}^{N} \mathcal{K}_{\cdot,i}^T \mathcal{K}'_{\cdot,i} + \sum_{j\neq i}\left(\mathcal{K}_{\cdot,i}^T \mathcal{K}_{\cdot,j}\right)^2 = -2\left(\sum_{i=1}^{N} \mathcal{K}_{\cdot,i}^T \mathcal{K}'_{\cdot,i}\right) + \|\mathcal{K}^T\mathcal{K} - \text{diag}(\mathcal{K}^T\mathcal{K})\|_F^2. \quad (18)$$

The normalization that is employed is to project all embeddings on a ball of radius $\mu$. This means that if their norm is lower than $\mu$, nothing will happen to them.

**Barlow Twins:** We consider that $\mathcal{K}$ is l2 normalized row-wise, i.e. dimensions are normalized. This gives us:

$$\mathcal{L}_{BT} = \sum_{j=1}^{M}\left(1 - (\mathcal{K}\mathcal{K}'^T)_{j,j}\right)^2 + \lambda\sum_{i,j,i\neq j}^{M}(\mathcal{K}\mathcal{K}'^T)_{j,i}^2 = \sum_{j=1}^{M}\left(1 - (\mathcal{K}\mathcal{K}'^T)_{j,j}\right)^2 + \lambda\|\mathcal{K}\mathcal{K}'^T - \text{diag}(\mathcal{K}\mathcal{K}'^T)\|_F^2. \quad (19)$$

**VICReg:** VICReg's criterion is defined as

$$\mathcal{L}_{VICReg} = \lambda\sum_{i=1}^{N}\|\mathcal{K}_{\cdot,i} - \mathcal{K}'_{\cdot,i}\|_2^2 + \mu\left(v(\mathcal{K}) + v(\mathcal{K}')\right) + \nu\left(c(\mathcal{K}) + c(\mathcal{K}')\right). \quad (20)$$

With $c$ a criterion that penalizes the off-diagonal terms of the covariance matrix as

$$c(\mathcal{K}) = \sum_{i\neq j}\text{Cov}(\mathcal{K})_{i,j}^2 = \|\mathcal{K}\mathcal{K}^T - \text{diag}(\mathcal{K}\mathcal{K}^T)\|_F^2 = L_{nc}, \quad (21)$$

and $v$ a criterion that aims at normalizing dimensions, i.e. rows of $\mathcal{K}$.

**TCR:** TCR's cost function is defined as

$$\mathcal{L}_{TCR} = -\frac{1}{2}\log\det\left(I + \alpha\text{Cov}(\mathcal{K})\right) = -\frac{1}{2}\log\det\left(I + \alpha\mathcal{K}\mathcal{K}^T\right) = -\frac{1}{2}\sum_{i}\log\left(1 + \alpha\sigma_i^2\right), \quad (22)$$

where $\sigma_i$ is the $i$-th singular value of $\mathcal{K}$.

# B    LINKS BETWEEN METHODS AND OUR CRITERIA

While we focus on methods for which the regularization is obtained through the criterion, several other methods can be linked informally to our results. The difficulty in linking them to $L_c$ or $L_{nc}$ can also come from choices that are motivated by practical limitations, such as the use of a memory bank, and which do not change methods fundamentally.

One of the most surprising lines of works, BYOL (Grill et al., 2020) and SimSiam (Chen & He, 2020), showed that using stop-gradient on one side of the encoder and using a predictor network to create asymmetry was enough to avoid collapse and learn good representations. Even though they do not avoid collapse explicitly through their criteria, recent works such as Halvagal et al. (2022) or Theorem 3 from Tian et al. (2021) have shown links between the training dynamics of SimSiam and variance and covariance regularization, akin to what $L_{nc}$ would lead to. While these analyses require assumptions such as the linearity of the encoder, they still help shine a light on SimSiam and BYOL's behavior and enable us to see how they can be related to our results.

Due to the popularity of sample-contrastive methods, several variants have emerged to improve their sample efficiency or their performance in general. One such modification is illustrated in MoCo (He et al., 2020; Chen et al., 2020b; 2021b) where a memory bank of sample is combined with an exponential moving average (EMA) of the encoder to provide better negative pairs and thus improve training. While this makes it hard to relate MoCo to our framework, it still relies on an InfoNCE criterion like SimCLR and thus leads to similar representations. SimCLR and MoCo become especially close near convergence since the online network and the EMA one will be very similar and thus the two methods also become more alike.

Clustering methods such as DeepCluster (Caron et al., 2018), SwAV (Caron et al., 2020) or DINO Caron et al. (2021) can also be related informally to sample-contrastive approaches. Similarly to MoCo, the main difference lies in the construction of the negative pairs, which are constructed using cluster centers here. The embeddings are then contrasted with these clustering prototypes using losses akin to InfoNCE. In DINO, the clustering aspect is more subtle as it is done online, thanks to the last linear layer of the projector which can be thought of as the bank of cluster prototypes, and the embeddings are then the outputs of the penultimate layer. Its projector can thus be decomposed into two parts, the first being the classical projector which is followed by L2 normalization, and the last layer which acts as a clustering layer thanks to the softmax that follows it. As such, while clustering methods cannot be clearly linked to our framework, a link to sample-contrastive methods is still present, even if only informally.

Overall, while not all methods can fit clearly into our results, we are still able to relate most of them to sample-contrastive or dimension-contrastive methods, even if it is with less rigor. This further reinforces the similarity between methods.

# C    PROOFS

**Lemma C.1.** Let $X, Y \sim \sigma^{D-1}$ two i.i.d. random variables corresponding to vectors uniformly distributed on $S^{D-1}$. Their dot product follows the following distribution

$$\frac{X^T Y + 1}{2} \sim \text{Beta}\left(\frac{D-1}{2}, \frac{D-1}{2}\right).$$

*Proof.* A similar result was proved in Fernandez et al. (2022), though we go one step further and derive the distribution of $\frac{X^T Y + 1}{2}$. We follow a more geometrical argument and invite the reader to confer Fernandez et al. (2022) for an alternative approach.

By the symmetry of the hypersphere, the distribution of $X^T Y$ is the same as the one of $X^T(1, 0 \ldots, 0)$, which corresponds to rotating the reference frame. The cumulative distribution function then corresponds to the surface of the hyperspherical cap of angle $\cos^{-1}(X_1)$.
Using the formulas for the area of a spherical cap on $S^D$ derived in Li (2011), as well as the fact than $\sin^2(\cos^{-1}(x)) = 1 - x^2$ we directly obtain that for $X^T Y > 0$ (i.e. $\cos^{-1}(X_1) \leq \frac{\pi}{2}$), we have $1 - (X^T Y)^2 \sim \text{Beta}\left(\frac{D-1}{2}, \frac{1}{2}\right)$.

Since the density of the Beta distribution has reflectional symmetry, we see that $(X^T Y)^2 \sim$ Beta $\left(\frac{1}{2}, \frac{D-1}{2}\right)$.

By substituting in $u = \frac{X^T Y + 1}{2}$ if follows directly that

$$u \sim \text{Beta}\left(\frac{D-1}{2}, \frac{D-1}{2}\right), \tag{23}$$

concluding the proof. $\qquad \square$

**Proposition C.2.** Considering an infinite amount of available negative samples, SimCLR and DCL's criteria lead to embeddings where for negative pairs $(x, x^-) \in \mathbb{R}^M$ we have

$$\mathbb{E}\left[x^T x^-\right] = 0 \quad \text{and} \quad \text{Var}\left[x^T x^-\right] = \frac{1}{M}. \tag{24}$$

*Proof.* The proof hinges on Theorem 1 from Wang & Isola (2020), which states that as the number of negative samples goes to infinity, optimizing the repulsive force of the InfoNCE criterion leads to uniformly distributed embeddings on the $M$-hypersphere.

This uniform distribution allows us to leverage Lemma C.1 in saying that as the number of negative samples goes to infinity, for any pair of random embeddings $X, Y$, we have $\frac{X^T Y + 1}{2} \sim$ Beta $\left(\frac{M-1}{2}, \frac{M-1}{2}\right)$.
We can directly obtain the two following properties

$$\mathbb{E}\left[\frac{X^T Y + 1}{2}\right] = \frac{\frac{M-1}{2}}{\frac{M-1}{2} + \frac{M-1}{2}} = \frac{1}{2} \Rightarrow \mathbb{E}\left[X^T Y\right] = 0, \tag{25}$$

$$\text{Var}\left[\frac{X^T Y + 1}{2}\right] = \frac{\frac{M-1}{2} \times \frac{M-1}{2}}{\left(\frac{M-1}{2} + \frac{M-1}{2}\right)^2 \left(\frac{M-1}{2} + \frac{M-1}{2} + 1\right)} = \frac{1}{4M} \Rightarrow \text{Var}\left[X^T Y\right] = \frac{1}{M}, \tag{26}$$

concluding the proof.

$$\square$$

**Proposition C.3.** SimCLR-abs/sq, DCL-sq/abs, as well as Spectral Contrastive Loss are sample-contrastive methods. Barlow Twins, VICReg, and TCR are dimension-contrastive methods.

*Proof.* **DCL-sq/abs:** We first take a look at DCL-sq/abs's criteria. We consider that $\mathcal{K}$ is l2 normalized column-wise, i.e. embeddings are normalized. Let $f : \mathbb{R} \to \mathbb{R}^+$ be either defined as $f(x) = x^2$ for DCL-sq or as $f(x) = |x|$ for DCL-abs. We have

$$\mathcal{L}_{\text{DCL}} = \sum_{i=1}^{N} -\log\left(\frac{e^{f\left(\mathcal{K}_{\cdot,i}^T \mathcal{K}_{\cdot,i}'\right)/\tau}}{\sum_{j \neq i} e^{f\left(\mathcal{K}_{\cdot,i}^T \mathcal{K}_{\cdot,j}\right)/\tau}}\right) = \sum_{i=1}^{N} -\frac{f\left(\mathcal{K}_{\cdot,i}^T \mathcal{K}_{\cdot,i}'\right)}{\tau} + \log\left(\sum_{j \neq i} e^{f\left(\mathcal{K}_{\cdot,i}^T \mathcal{K}_{\cdot,j}\right)/\tau}\right). \tag{27}$$

The first part of this criterion is the invariance criterion and the second part is the LogSumExp($LSE$) of embeddings' similarity. We know that this is a smooth approximation of the max operator with the following bounds:

$$\max\left(\{\forall j \neq i, \ f\left(\mathcal{K}_{\cdot,i}^T \mathcal{K}_{\cdot,j}\right)\}\right) \leq \tau \log\left(\sum_{j \neq i} e^{f\left(\mathcal{K}_{\cdot,i}^T \mathcal{K}_{\cdot,j}\right)/\tau}\right) \leq \max\left(\{\forall j \neq i, \ f\left(\mathcal{K}_{\cdot,i}^T \mathcal{K}_{\cdot,j}\right)\}\right) + \tau \log(N-1). \tag{28}$$

We can thus say that using either

$$\sum_{i=1}^{N} \log\left(\sum_{j \neq i} e^{f\left(\mathcal{K}_{\cdot,i}^T \mathcal{K}_{\cdot,j}\right)/\tau}\right) \quad \text{or} \quad \sum_{i=1}^{N} \max_{j \neq i} f\left(\mathcal{K}_{\cdot,i}^T \mathcal{K}_{\cdot,j}\right), \tag{29}$$

as repulsive force will lead to the same result, a diagonal Gram matrix. Since this is the same goal as for our sample-contrastive criterion, DCL-sq and DCL-abs are sample-contrastive methods.

The link to $L_c$ is more visible with the right term, which corresponds to only penalizing one value per row/column of the Gram matrix. While this is less effective than penalizing all of them at once, given sufficient training iterations it will converge to the same solution.

**SimCLR-sq/abs:** We now take a look at SimCLR-abs/sq's criteria. We consider that $\mathcal{K}$ is l2 normalized column-wise, i.e. embeddings are normalized. Let $f : \mathbb{R} \rightarrow \mathbb{R}^+$ be either defined as $f(x) = x^2$ for SimCLR-sq or as $f(x) = |x|$ for SimCLR-abs. We have

$$\mathcal{L}_{\text{SimCLR}} = \sum_{i=1}^{N} - \log \left( \frac{e^{f(\mathcal{K}_{\cdot,i}^T \mathcal{K}_{\cdot,i}')/\tau}}{e^{f(\mathcal{K}_{\cdot,i}^T \mathcal{K}_{\cdot,i}')/\tau} + \sum_{j \neq i} e^{f(\mathcal{K}_{\cdot,i}^T \mathcal{K}_{\cdot,j})/\tau}} \right) \tag{30}$$

$$= \sum_{i=1}^{N} -\frac{f\left(\mathcal{K}_{\cdot,i}^T \mathcal{K}_{\cdot,i}'\right)}{\tau} + \log \left( e^{f(\mathcal{K}_{\cdot,i}^T \mathcal{K}_{\cdot,i}')/\tau} + \sum_{j \neq i} e^{f(\mathcal{K}_{\cdot,i}^T \mathcal{K}_{\cdot,j})/\tau} \right). \tag{31}$$

Due to the presence of the positive pair in the repulsive force (right term), we cannot use the same reasoning with the max operator as for DCL-sq/abs which gave a clear intuition.

Nonetheless one can clearly see that to minimize this criterion, all the similarities between the negative pairs, i.e. $\forall i, \forall j \neq i,\ f\left(\mathcal{K}_{\cdot,i}^T \mathcal{K}_{\cdot,j}\right)$, need to be minimized. As this will result in a diagonal Gram matrix, we can say that minimizing this criterion will also minimize our sample-contrastive one. We can thus conclude that SimCLR-sq and SimCLR-abs are sample-contrastive methods.

**Spectral Constrastive Loss:** We will now consider Spectral Constrastive Learning's criterion. We have

$$\mathcal{L}_{SCL} = -2 \sum_{i=1}^{N} \mathcal{K}_{\cdot,i}^T \mathcal{K}_{\cdot,i}' + \sum_{j \neq i} \left(\mathcal{K}_{\cdot,i}^T \mathcal{K}_{\cdot,j}\right)^2 = -2 \left( \sum_{i=1}^{N} \mathcal{K}_{\cdot,i}^T \mathcal{K}_{\cdot,i}' \right) + \|\mathcal{K}^T \mathcal{K} - \text{diag}(\mathcal{K}^T \mathcal{K})\|_F^2. \tag{32}$$

This means that Spectral Constrastive Loss also falls in the sample-contrastive category.

**Barlow Twins:** Looking at Barlow Twin's criterion we have

$$\mathcal{L}_{BT} = \sum_{j=1}^{M} \left(1 - (\mathcal{K}\mathcal{K}'^T)_{j,j}\right)^2 + \lambda \sum_{i,j,i \neq j}^{M} (\mathcal{K}\mathcal{K}'^T)_{j,i}^2 = \sum_{j=1}^{M} \left(1 - (\mathcal{K}\mathcal{K}'^T)_{j,j}\right)^2 + \lambda \|\mathcal{K}\mathcal{K}'^T - \text{diag}(\mathcal{K}\mathcal{K}'^T)\|_F^2. \tag{33}$$

Since the distribution of augmentations is the same for both views of the images, and the backbone is shared, taking a negative pair from $\mathcal{K}$ or $\mathcal{K}'$ is the same. Barlow Twins' criterion can then be rewritten as

$$\mathcal{L}_{BT} = \sum_{j=1}^{M} \left(1 - (\mathcal{K}\mathcal{K}'^T)_{j,j}\right)^2 + \lambda \|\mathcal{K}\mathcal{K}^T - \text{diag}(\mathcal{K}\mathcal{K}^T)\|_F^2. \tag{34}$$

As such the right part of Barlow Twins' criterion is indeed the dimension-contrastive criterion, making Barlow Twins a dimension-contrastive method.

**VICReg:** VICReg's criterion is defined as

$$\mathcal{L}_{VICReg} = \lambda \sum_{i=1}^{N} \|\mathcal{K}_{\cdot,i} - \mathcal{K}_{\cdot,i}'\|_2^2 + \mu \left(v(\mathcal{K}) + v(\mathcal{K}')\right) + \nu \left(c(\mathcal{K}) + c(\mathcal{K}')\right). \tag{35}$$

Recall that $c$ is a criterion that penalizes the off-diagonal terms of the covariance matrix as follows:

$$c(\mathcal{K}) = \sum_{i \neq j} \text{Cov}(\mathcal{K})_{i,j}^2 = \|\mathcal{K}\mathcal{K}^T - \text{diag}(\mathcal{K}\mathcal{K}^T)\|_F^2 = L_{nc}. \tag{36}$$

This means that VICReg is a dimension-contrastive method.

**TCR:** TCR's cost function is defined as

$$\mathcal{L}_{TCR} = -\frac{1}{2}\log\det\left(I + \alpha\mathrm{Cov}(\mathcal{K})\right) = -\frac{1}{2}\log\det\left(I + \alpha\mathcal{K}\mathcal{K}^T\right) = -\frac{1}{2}\sum_i \log\left(1 + \alpha\sigma_i^2\right),$$
(37)

where $\sigma_i$ is the $i$-th singular value of $\mathcal{K}$. As discussed in Li et al. (2022b), this criterion leads to a diagonal covariance matrix, similarly to the non-contrastive criterion. We can thus say using either

$$-\frac{1}{2}\sum_i \log\left(1 + \alpha\sigma_i^2\right) \quad \text{or} \quad \|\mathcal{K}\mathcal{K}^T - \mathrm{diag}(\mathcal{K}\mathcal{K}^T)\|_F^2$$
(38)

will lead to diagonal covariance matrices, or similarly, null off-diagonal terms in the Covariance matrix. This means that TCR also falls in the category of dimension-contrastive methods. □

**Theorem C.4.** The sample-contrastive and dimension-contrastive criteria $L_c$ and $L_{nc}$ are equivalent up to row and column normalization of the embedding matrix $\mathcal{K}$. Consider a batch size of $N$ and an embedding dimension of $M$. We have:

$$L_{nc} + \sum_{j=1}^{M} \|\mathcal{K}_{j,\cdot}\|_2^4 = L_c + \sum_{i=1}^{N} \|\mathcal{K}_{\cdot,i}\|_2^4.$$
(39)

*Proof.* This proof is heavily inspired by the proof of Lemma 3.2 from Le et al. (2011) which provides a similar result for doubly stochastic matrices.
We have

$$L_{nc} = \|\mathcal{K}\mathcal{K}^T - \mathrm{diag}(\mathcal{K}\mathcal{K}^T)\|_F^2$$
(40)

$$= tr\left[(\mathcal{K}\mathcal{K}^T - \mathrm{diag}(\mathcal{K}\mathcal{K}^T))^T(\mathcal{K}\mathcal{K}^T - \mathrm{diag}(\mathcal{K}\mathcal{K}^T))\right]$$
(41)

$$= tr(\mathcal{K}\mathcal{K}^T\mathcal{K}\mathcal{K}^T) - 2tr(\mathcal{K}\mathcal{K}^T \mathrm{diag}(\mathcal{K}\mathcal{K}^T)) + tr(\mathrm{diag}(\mathcal{K}\mathcal{K}^T) \mathrm{diag}(\mathcal{K}\mathcal{K}^T))$$
(42)

$$= tr(\mathcal{K}\mathcal{K}^T\mathcal{K}\mathcal{K}^T) - tr(\mathcal{K}\mathcal{K}^T \mathrm{diag}(\mathcal{K}\mathcal{K}^T))$$
(43)

$$= tr(\mathcal{K}^T\mathcal{K}\mathcal{K}^T\mathcal{K}) - tr(\mathcal{K}\mathcal{K}^T \mathrm{diag}(\mathcal{K}\mathcal{K}^T)).$$
(44)

Similarly for $L_c$, we obtain

$$L_c = \|\mathcal{K}^T\mathcal{K} - \mathrm{diag}(\mathcal{K}^T\mathcal{K})\|_F^2$$
(45)

$$= tr(\mathcal{K}^T\mathcal{K}\mathcal{K}^T\mathcal{K}) - tr(\mathcal{K}^T\mathcal{K} \mathrm{diag}(\mathcal{K}^T\mathcal{K})).$$
(46)

Since $(\mathcal{K}^T\mathcal{K})_{i,i} = \|\mathcal{K}_{\cdot,i}\|_2^2$ we deduce that $tr(\mathcal{K}^T\mathcal{K} \mathrm{diag}(\mathcal{K}^T\mathcal{K})) = \sum_{i=1}^{N} \|K_{\cdot,i}\|_2^4$. Similarly, we obtain that $tr(\mathcal{K}\mathcal{K}^T \mathrm{diag}(\mathcal{K}\mathcal{K}^T)) = \sum_{j=1}^{M} \|K_{j,\cdot}\|_2^4$.
Plugging this back in, we finally deduce that

$$L_{nc} = L_c + \sum_{i=1}^{N} \|\mathcal{K}_{\cdot,i}\|_2^4 - \sum_{j=1}^{M} \|\mathcal{K}_{j,\cdot}\|_2^4,$$
(47)

concluding the proof.

□

**Lemma C.5.** If embeddings are normalized such that $\forall i$, $\|\mathcal{K}_{\cdot,i}\|_2 = a$ we have

$$\frac{N^2}{M}a^4 \leq \sum_{j=1}^{M} \|\mathcal{K}_{j,\cdot}\|_2^4 \leq N^2 a^4.$$
(48)

Conversely, if dimensions are normalized such that $\forall j$, $\|\mathcal{K}_{j,\cdot}\|_2 = a$ we have

$$\frac{M^2}{N}a^4 \leq \sum_{i=1}^{N} \|\mathcal{K}_{\cdot,i}\|_2^4 \leq M^2 a^4.$$
(49)

*Proof.* We start with the first set of inequalities. Since $\forall i,\ \|\mathcal{K}_{i,\cdot}\|_2^2 \geq 0$ we have

$$\sum_{j=1}^{M} \|\mathcal{K}_{j,\cdot}\|_2^4 \leq \left( \sum_{j=1}^{M} \|\mathcal{K}_{j,\cdot}\|_2^2 \right)^2 = \|\mathcal{K}\|_F^4 = N^2 a^4. \tag{50}$$

Which gives us our upper bound. For the lower bound, using the convexity of the function $f : x \to x^2$ we obtain

$$\frac{1}{M} \sum_{j=1}^{M} \|\mathcal{K}_{j,\cdot}\|_2^4 \geq \left( \frac{1}{M} \sum_{j=1}^{M} \|\mathcal{K}_{j,\cdot}\|_2^2 \right)^2 = \frac{N^2}{M^2} a^4. \tag{51}$$

Combining those two inequalities gives us the desired bounds.

For the second set of inequalities, we follow the same reasoning and use the fact that in this scenario $\|\mathcal{K}\|_F^2 = M a^2$ giving us the aforementioned bounds and concluding the proof.

$\square$

## D  TRAINING PROCEDURE

For training, we follow common procedure and use a ResNet-50 backbone (He et al., 2016), with the LARS (You et al., 2017) optimizer. We use by default a base learning rate of $0.3$ and compute the effective learning rate as $lr = base\_lr \times \frac{batch\_size}{256}$. We also use a momentum of $0.9$ and weight decay of $10^{-6}$. The learning rate follows a cosine annealing schedule after a 10-epoch linear warmup. We train for 100 epochs in all of our experiments.
For data augmentation, we follow the protocol of BYOL (Grill et al., 2020) which is as follows

Table S1: Image augmentation parameters, taken from (Grill et al., 2020).

| Parameter | View 1 | View 2 |
|---|---|---|
| Random crop probability | 1.0 | 1.0 |
| Horizontal flip probability | 0.5 | 0.5 |
| Color jittering probability | 0.8 | 0.8 |
| Brightness adjustment max intensity | 0.4 | 0.4 |
| Contrast adjustment max intensity | 0.4 | 0.4 |
| Saturation adjustment max intensity | 0.2 | 0.2 |
| Hue adjustment max intensity | 0.1 | 0.1 |
| Grayscale probability | 0.2 | 0.2 |
| Gaussian blurring probability | 1.0 | 0.1 |
| Solarization probability. | 0.0 | 0.2 |

Each experiment was run on $8$ Nvidia V100 GPUs, with 32GB of memory each, and took around $24$ hours to complete.

While this was our base experimental protocol, it was adapted for each method, mostly by changing method-specific hyperparameters as well as the learning rate, confer supplementary section K for the exact hyperparameters used for each experiment. The Pytorch pseudocode for VICReg-exp and VICReg-ctr is also available in supplementary section L.

## E  ONLINE LINEAR PROBE

As previously discussed, to evaluate our experiments, we relied on the use of a linear classifier that is trained jointly with our main network. This means that it is trained on suboptimal representations and stronger augmentations compared to what is typically done for linear evaluation. Even though these two approaches seem closely related, we are interested in finding how well they are correlated.

To do so, we trained a linear evaluation on VICReg and VICreg-exp with a projector architecture of $8192 - 8192 - d$, $d \in [256, 512, 1024, 2048, 8192]$ using the following protocol. We train the linear classifier on frozen representations for 100 epochs with a batch size of 1024 using the SGD optimizer with a base learning rate $0.25$ (for VICReg) or $1.4$ (for VICReg-exp), momentum $0.9$, weight decay $10^{-6}$ and using a cosine annealing learning rate scheduler. We compute the learning rate as $lr = base\_lr \times \frac{batch\_size}{256}$. For augmentations, we follow standard procedure and use random cropping with a scale between $0.08$ and $1$ with an image size of $224 \times 224$ and horizontal flip with a probability $0.5$ during training. For evaluation, we do a center crop.

Table S2: Relationship in performance between the online linear probe and the offline linear classifier. We used VICReg and an expander with architecture $8192 - 8192 - d$.

| Embedding dimension | 256 | 512 | 1024 | 2048 | 8192 |
|---|---|---|---|---|---|
| Online top-1 | 65.01 | 66.72 | 68.06 | 68.06 | 68.13 |
| Offline top-1 | 65.11 | 66.64 | 67.96 | 68.00 | 68.02 |

Table S3: Relationship in performance between the online linear probe and the offline linear classifier. We used VICReg-exp and an expander with architecture $8192 - 8192 - d$.

| Embedding dimension | 256 | 512 | 1024 | 2048 | 8192 |
|---|---|---|---|---|---|
| Online top-1 | 65.24 | 66.71 | 67.86 | 68.00 | 67.93 |
| Offline top-1 | 65.30 | 66.58 | 67.83 | 67.89 | 68.18 |

As we can see in table S2 and S3, the performance achieved by the offline classifier is extremely close to the performance of the online classifier. While the online classifier cost in compute is negligible, the linear evaluation is almost as long as the pretraining due to data loading bottlenecks and it requires a significant amount of learning rate tuning. This makes this online classifier a very appealing alternative since it demonstrates very correlated performances for a fraction of the computing cost.

Training a linear regression on those two sets of evaluations gives a model with a slope of $0.97$, an intercept of $2.1$, and an $R^2$ of $1.0$. It is worth noting that since most values are close to $68$, the fitting of linear regression on this data is sensitive to noise. Nonetheless, the low intercept, as well as the closeness of the slope to $1$, confirm the negligible gap between the two evaluation methods that we previously intuited.

## F    ADDITIONAL EVIDENCE OF THE SIMILARITY OF LEARNED REPRESENTATIONS

The goal of this section is to provide additional empirical evidence of the similar properties of representations learned by sample-contrastive and dimension-contrastive methods. To this effect, we will evaluate representations with a $k$-nn classifier and compare their similarities with CKA (Kornblith et al., 2019).

$k$**-nn evaluation.** In order to see if our previous results only validated similar performance in a linear classification setting, we will look at performance with $k$-nn classifiers which evaluate how well a metric is preserved instead of linear separability. We rely on the protocol of Bardes et al. (2021), and use values of $k$ in $[1, 5, 10, 20, 50, 200]$, with temperatures in $[0.05, 0.07, 0.1, 0.2, 0.5, 1]$ for the weighting of the classifiers. We then look at the best performance achieved by all methods to give a comparison that is as fair as possible.

As we can see in figure S1, we are able to retrieve behaviors similar to figure 1, although results appear less stable for VICReg-exp and VICReg-ctr. Nonetheless, looking at the transition VICReg-exp $\rightarrow$ VICReg-ctr we can see that the peak performance is still preserved, further validation our results for these methods were the dimension-contrastive and sample-contrastive natures are isolated. Similarly as for linear evaluation, the original implementation of SimCLR performs significantly worse than other methods, but our tuned SimCLR can recover the performance of VICReg with a 5.5 point

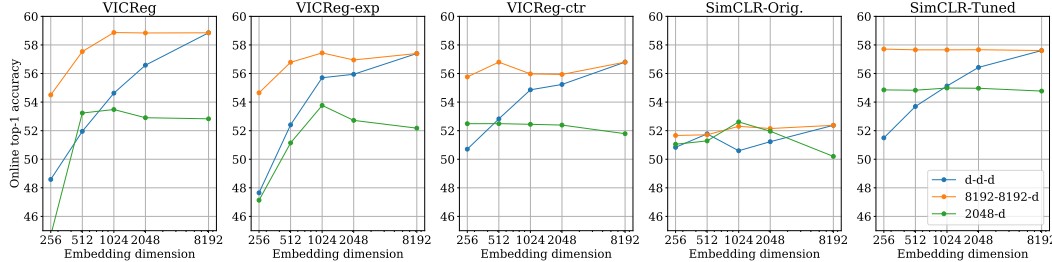

Figure S1: Reproduction of figure 1 with a $k$-nn classifier. We notice the same pattern as previously, where going from dimension-contrastive to sample-contrastive does not lead to a significant drop in performance.

increase in performance. This highlights how the practical implications of our results extend beyond linear classification, while further validating our theory.

**CKA.** CKA (Centered Kernel Alignment) (Kornblith et al., 2019) is a powerful tool to study the similarities between representations, which relies on HSIC (Hilbert-Schmidt Independance Criterion) (Gretton et al., 2005) with a given kernel. We will use a linear kernel for simplicity. For each method, we will study three different experiments that reached the same level of performance to measure both intra- and inter-method correlation between representtions. We also consider a random network to give a lower bound of what we can expect.

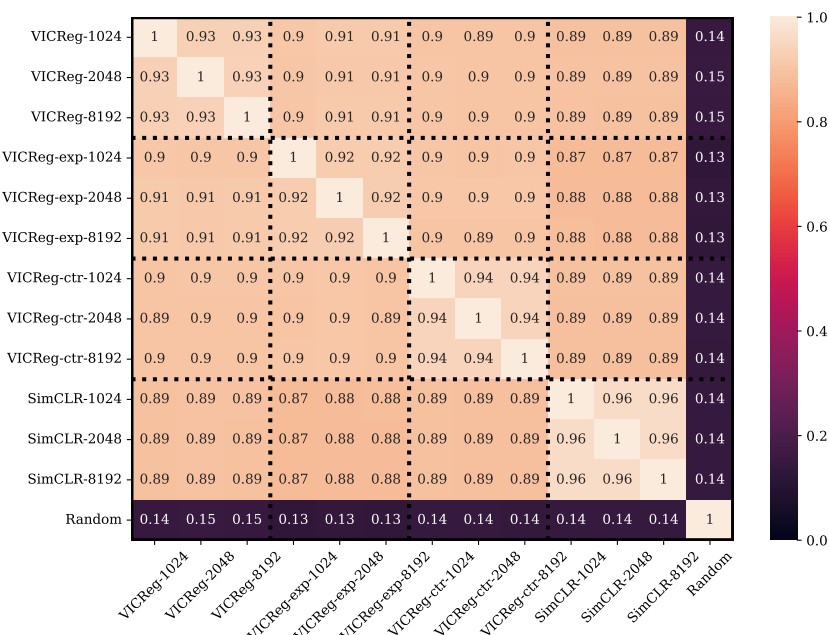

Figure S2: CKA of the learned representations on all considered methods. For a given method, we use experiments with different output dimensions (1024,2048,8192) that achieved equivalent performance.

As we can see in figure S2, all of the learned representations are highly correlated, where intra- and inter-methods CKA are very similar. This both shows that different self-supervised methods, whether dimension-contrastive or sample-contrastive, provide consistent representations over different runs and also all learn similar representations. These results contrast with the findings in Figure 2 from Gwilliam & Shrivastava (2022) where they found that different methods lead to representations with low CKA. We believe that their findings can be explained by different training setups between methods since the models used were trained with different projectors and data augmentation.

Both of these analyses help cement our results, where we can now say that through the lens of linear classification, $k$-nn classification, and CKA, all studied self-supervised methods produce extremely similar representations.

## G  IMPACT OF THE SIMILARITY MEASURE ON SIMCLR

While SimCLR uses cosine similarity to push away negative pairs, we will look at what happens when we use the square or absolute value of cosine similarities, as in SimCLR-sq or SimCLR-abs.

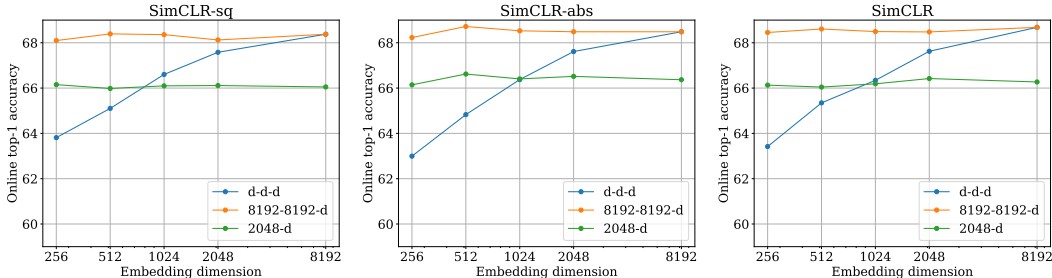

Figure S3: Influence of using squared or absolute values of the cosine similarities for VICReg, with different projector architectures.

As we can see in figure S3, the use of the squared or absolute values of the similarities did not impact the performance on image classification, it even improved slightly with a large projector when using the absolute values, achieving $68.7\%$ top-1 accuracy.

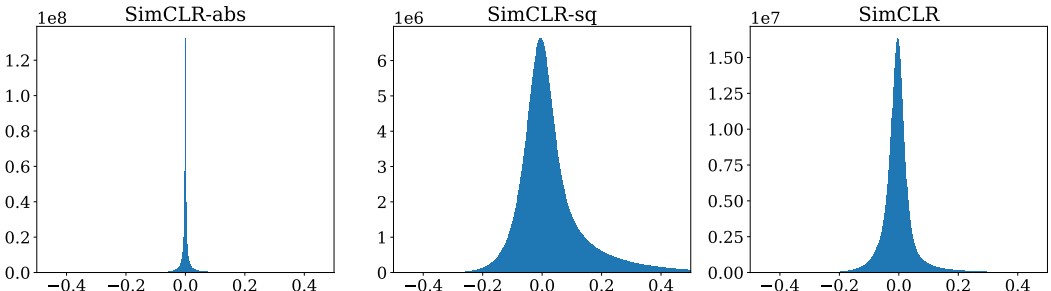

Figure S4: Histogram of cosine similarities for negative pairs in SimCLR-abs, SimCLR-sq and SimCLR.

As we can see in figure S4, for all three methods we obtain a distribution of cosine similarities that is centered at 0, but they all have very different standard deviations. The main culprit of this difference is dimensional collapse, as studied extensively in Jing et al. (2022). We study this behavior in figure S5, where we see that the three methods show different levels of collapse. While SimCLR-abs appears to have an almost full rank embedding matrix, we can see some collapse at around 256 dimensions for SimCLR, and 64 for SimCLR-sq. Per proposition 3.1, we know that with a perfect optimization of SimCLR's criterion, we should observe a variance of $1/D$ for the cosine similarities, if we have $D$-dimensional embeddings. However this is not the ambient dimension but the embeddings' dimension, and so when combining this result with the dimensional collapse, we clearly see that SimCLR-abs should have less variance as it has the least amount of collapse, and SimCLR-sq the highest variance as it has the most amount of collapse. Since this is what we observe in practice, these results are coherent with the three methods producing similar cosine similarities distributions, albeit with different standard deviations depending on the amount of dimensional collapse.

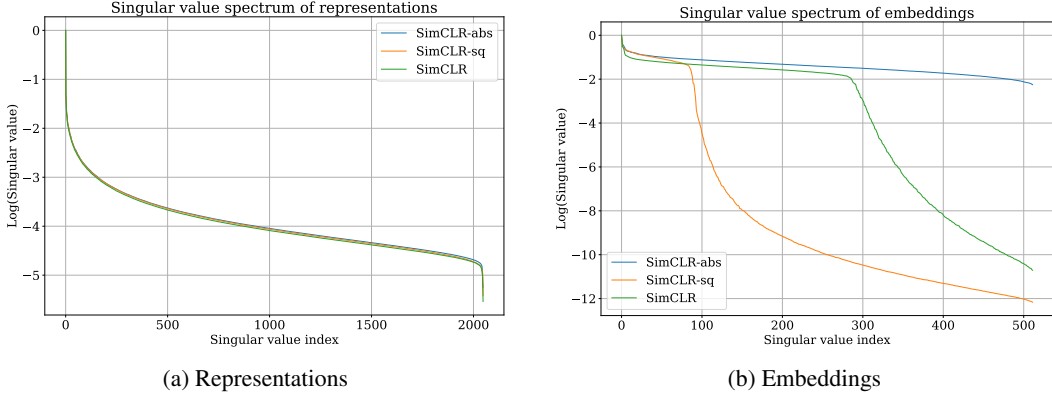

(a) Representations                                    (b) Embeddings

Figure S5: Singular value distribution of the embeddings and representations computed on the training set of ImageNet for SimCLR, SimCLR-abs and SimCLR-sq. All methods use 512 dimensional embeddings.

## H    ROW AND COLUMN NORMS INTERPLAY

While we provided bounds that apply to any matrix in lemma 3.4, in practice embedding matrices have a particular structure and one can wonder where the norms are in between the relatively distant bounds.
To study this we took 1024 images from ImageNet, computed the corresponding embedding matrices, and then l2-normalized the rows or columns.

Table S4: The empirical interplay between embedding matrix norms under row- or column-wise l2-normalization for different methods and projector architectures. We abbreviate thousands with k and millions with M. The experiment "Random" indicates a randomly initialized network.

| Experiment | Projector | Colum normalization | | | Row normalization | | |
|---|---|---|---|---|---|---|---|
| | | $\frac{N^2}{M}$ | $\sum \|\mathcal{K}_{j,\cdot}\|_2^4$ | $N^2$ | $\frac{M^2}{N}$ | $\sum \|\mathcal{K}_{\cdot,i}\|_2^4$ | $M^2$ |
| VICReg | $8192 - 8192 - 8192$ | 128 | 128.19 | 1M | 65k | 83k | 67M |
| VICReg-exp | $8192 - 8192 - 8192$ | 128 | 128.26 | 1M | 65k | 95k | 67M |
| VICReg-ctr | $8192 - 8192 - 512$ | 2048 | 2078 | 1M | 256 | 287 | 262k |
| SimCLR | $8192 - 8192 - 512$ | 2048 | 2061 | 1M | 256 | 433.54 | 262k |
| | $8192 - 8192 - 8192$ | 128 | 129.43 | 1M | 65k | 113k | 67M |
| Random | $8192 - 8192 - 8192$ | 128 | 361.34 | 1M | 65k | 75k | 67M |

As we can see in table S4, for every method, in any expansion or projection scenario, we are always close to the lower bound, deviating by a factor of 3 at most. This is significantly smaller than the factors $N$ or $M$ in lemma 3.4 which are tight when making no assumptions on the embedding matrix $\mathcal{K}$. As previously discussed these extreme cases consist respectively of a constant matrix and one with only one non-zero element per row/column. It is logical that the embedding matrices that we have in practice are closer to a constant matrix, with a uniform spread of information, even though they still present some sparsity.
As such, for all practical concerns, the bounds are much closer in practice than they theoretically are. This means that the sample-contrastive and dimension-contrastive criteria will also be closer in practice.

# I  IMPACT OF THE PROJECTOR CAPACITY

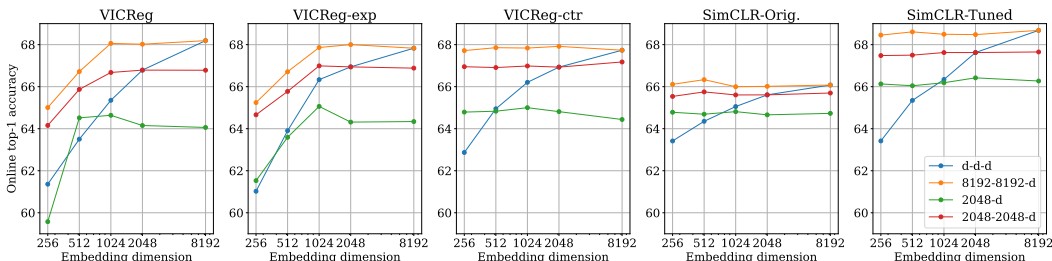

Figure S6: Online performance on ImageNet for VICReg, VICReg-exp, VICReg-ctr, and SimCLR with respect to embedding dimensions when changing the projector's architecture.

As discussed in section 5, the design of the projector plays a significant role in downstream performance. In figure S6, we also overlay the results for a projector with architecture $2048 - 2048 - d$ on top of the previously discussed ones. Such a projector offers similar behavior as an $8192 - 8192 - d$ one, but with a bit lower performance. The drop in performance is especially noticeable in dimension-contrastive methods.

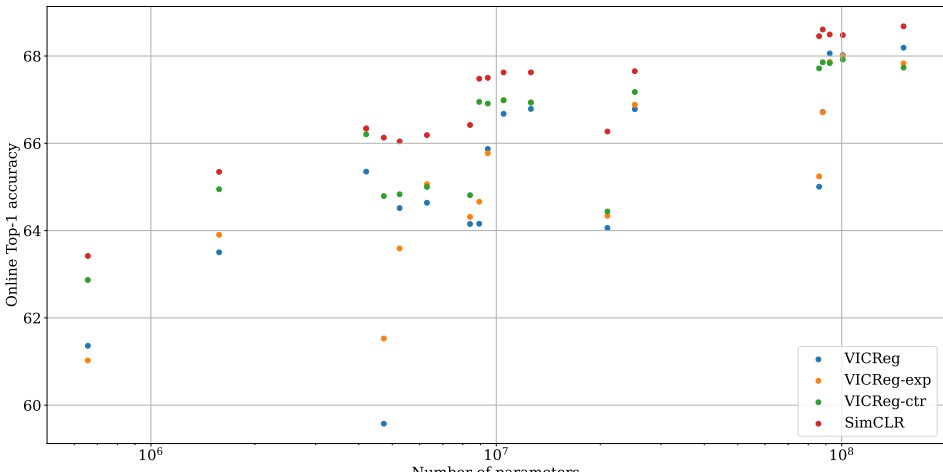

Figure S7: Online performance on ImageNet for VICReg, VICReg-exp, VICReg-ctr, and SimCLR with respect to the number of parameters in their projector.

As we can see in figure S7, if we take a look at the performance with respect to the number of parameters of the projector we can see a clear trend that indicates that performance is increased when increasing the number of parameters of the projector. This conclusion holds for all methods though there are some scenarios that are clear outliers. For example, for VICReg and VICReg exp we can see that with a $2048 - 256$ projector, the performance is significantly lower than expected.

While it would be interesting to see if this increase in performance saturates at some point, our largest projectors already have 151 million parameters. Increasing it further quickly starts to become impractical due to memory constraints during training, and as such, we leave this study to future work.

Another aspect worth mentioning is that the increase in performance when increasing the number of parameters is not automatic. For example for VICReg, the scenario $2048 - 2048 - 1024$ achieves 66.68% top-1 for 10 million parameters, but the scenario $8192 - 8192 - 256$ only achieves 65.01% even though it has 86 million parameters. This drastic difference suggests that some care must be taken when designing the projector and that even though the number of parameters is important, the architecture in itself also is.

## J    INFLUENCE OF LOSS FUNCTION DESIGN ON OPTIMIZATION QUALITY

As previously discussed, the introduction of VICReg-exp allows us to study the influence of the use of the LogSumExp operator in the repulsive force, and VICReg-ctr to study the difference between sample-contrastive and dimension-contrastive methods when comparing it to VICReg-exp. This enables us to quantify the impact of these design choices on the quality of the optimization process.

While a perfect optimization of the aforementioned criteria would lead to embeddings with similar properties for the covariance and Gram matrix, one can wonder how well they are optimized in practice and whether design choices have a significant impact. To this effect we will look at the Gram and Covariance matrices after optimization, both on the embeddings to study the quality of the optimization process and on the representations to study the transferability of this process to the representations since they are used for downstream tasks. For the embeddings, we use the same normalization process as is used during training, and we center the representations to alleviate the fact that the last ReLU layer constrains them to the positive orthant. This centering on the representations is only done to make the visualization more interpretable.

As we can see in figure S8, while VICReg penalizes the off-diagonal terms of the covariance matrix and not the Gram matrix, both matrices have off-diagonal terms that are significantly smaller than their diagonal counterparts. Similarly for VICReg-exp, we can see that both the Gram and covariance matrices are dominated by their diagonal in the embedding space, though there is noise in the off-diagonal terms. This is due to the use of the LogSumExp, which as a smooth approximation of the max operator, will mostly penalize the largest values. On the other hand, using squared values will make them penalized by their absolute and not relative value.
We also observe the same behavior for VICReg-ctr and SimCLR, leading to Gram and covariance matrices that are dominated by their diagonal but are overall noisier than for VICReg and VICReg-exp. This suggests that the main culprit of this noise is indeed the LogSumExp but that the sample-contrastive nature of VICReg-ctr and SimCLR also played a role in creating it.

Looking at the representations, the differences between the methods start to fade. They all still produce Gram and covariance matrices that are dominated by their diagonal, but with some off-diagonal noise. Even though we could see a clear difference in the quality of the optimization in the embedding space, the similarity in the representation space makes it harder to interpret for practical scenarios. Indeed, we saw that all methods can be made to perform the same when evaluating the representations.

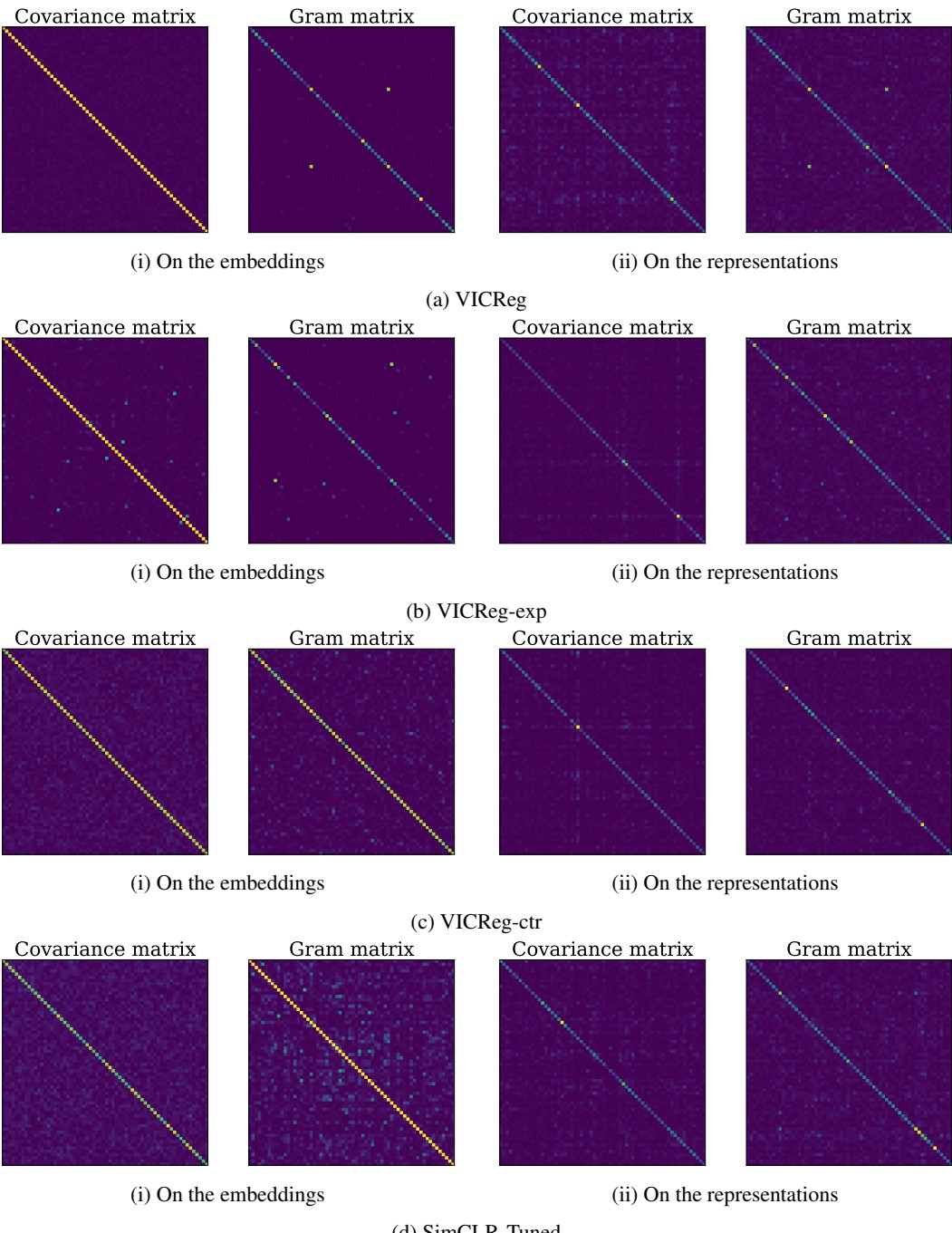

Figure S8: Covariance and similarity matrices on a random part of the ImageNet training set, using VICReg, VICReg-exp, VICReg-ctr, and SimCLR pretrained on ImageNet for 100 epochs. The covariance matrix is limited to the first 64 dimensions, while the Gram matrix is limited to the first 64 samples. In all cases, we used a projector with an output dimension of 2048, the same as the representation dimension.

# K COMPLETE PERFORMANCE AND HYPERPARAMETER TABLES

Table S5: Top-1 accuracy on ImageNet using the online linear classifier, including all performances for figures 1 and S6.

| Experiment | Projector | 256 | 512 | 1024 | 2048 | 8192 |
|---|---|---|---|---|---|---|
| VICReg | $d-d-d$ | 61.36 | 63.50 | 65.35 | 66.74 | 68.13 |
| | $8192-8192-d$ | 65.01 | 66.72 | 68.06 | 68.07 | 68.13 |
| | $2048-d$ | 59.57 | 64.51 | 64.64 | 64.15 | 64.06 |
| | $2048-2048-d$ | 64.16 | 65.87 | 66.68 | 66.74 | 66.78 |
| VICReg-exp | $d-d-d$ | 61.02 | 63.90 | 66.33 | 66.82 | 67.93 |
| | $8192-8192-d$ | 65.24 | 66.71 | 67.86 | 68.00 | 67.93 |
| | $2048-d$ | 61.53 | 63.59 | 65.06 | 64.31 | 64.34 |
| | $2048-2048-d$ | 64.66 | 65.77 | 66.99 | 66.82 | 66.88 |
| VICReg-ctr | $d-d-d$ | 62.87 | 64.95 | 66.21 | 66.93 | 67.73 |
| | $8192-8192-d$ | 67.72 | 67.86 | 67.84 | 67.92 | 67.73 |
| | $2048-d$ | 64.79 | 64.83 | 65.00 | 64.81 | 64.44 |
| | $2048-2048-d$ | 66.95 | 66.91 | 66.98 | 66.93 | 67.18 |
| SimCLR-Orig | $d-d-d$ | 63.42 | 64.35 | 65.05 | 65.61 | 66.08 |
| | $8192-8192-d$ | 66.11 | 66.33 | 66.00 | 66.02 | 66.08 |
| | $2048-d$ | 64.78 | 64.69 | 64.83 | 64.66 | 64.73 |
| | $2048-2048-d$ | 65.36 | 65.75 | 65.61 | 65.61 | 65.70 |
| SimCLR-Tuned | $d-d-d$ | 63.42 | 65.35 | 66.36 | 67.62 | 68.68 |
| | $8192-8192-d$ | 68.45 | 68.61 | 68.49 | 68.48 | 68.68 |
| | $2048-d$ | 66.13 | 66.04 | 66.19 | 66.42 | 66.27 |
| | $2048-2048-d$ | 67.48 | 67.50 | 67.62 | 67.62 | 67.65 |

Table S6: Hyperparameters used for the results in table S5. Sim., Var. and Cov. indicate the weights of the criteria in VICReg and its variations. $\tau$ indicates the temperature used for LogSumExp-based methods. The hyperparameters for VICReg and SimCLR are usable with the official implementations. For VICReg-exp and VICReg-ctr, the hyperparameters are compatible with the pseudocode in section L.

| Experiment | Projector | Batch size | base lr | VICReg | | | $\tau$ |
|---|---|---|---|---|---|---|---|
| | | | | Sim. | Var. | Cov. | |
| VICReg | $d=256$ | 1024 | 0.3 | 25 | 25 | 4 | |
| | $d=512$ | 1024 | 0.3 | 25 | 25 | 2 | |
| | $d=1024$ | 1024 | 0.3 | 25 | 25 | 2 | |
| | $d=2048$ | 1024 | 0.3 | 25 | 25 | 2 | |
| | $d=8192$ | 1024 | 0.3 | 25 | 25 | 0.5 | |
| VICReg-exp | $d \neq 8192$ | 1024 | 0.5 | 1 | 1 | 2 | 0.1 |
| | $d=8192$ | 1024 | 0.8 | 1 | 1 | 2 | 0.1 |
| VICReg-ctr | All | 1024 | 0.6 | 1 | 1 | 1 | 0.15 |
| SimCLR-Tuned | All | 2048 | 0.5 | | | | 0.15 |

## L VICReg VARIATIONS PSEUDOCODE

**Algorithm 1:** VICReg-exp PyTorch pseudocode.

```
# f: encoder network,p: projector network, lambda, mu, nu:
   coefficients of the invariance, variance and covariance
   losses, N: batch size, D: dimension of the
   representations, tau: temperature
# mse_loss: Mean square error loss function, relu: ReLU
   activation function, cut_out_diag: remove the diagonal of
    a matrix,

for x in loader: # load a batch with N samples
   # two randomly augmented versions of x
   x_a, x_b = augment(x)

   # compute embeddings
   k_a = p(f(x_a)) # N x D
   k_b = p(f(x_b)) # N x D

   # invariance loss
   sim_loss = mse_loss(k_a, k_b)

   # variance loss
   std_k_a = torch.sqrt(k_a.var(dim=0) + 1e-04)
   std_k_b = torch.sqrt(k_b.var(dim=0) + 1e-04)
   std_loss = torch.mean(relu(1 - std_k_a))/2 + torch.mean(
      relu(1 - std_k_b))/2

   # covariance loss
   k_a = k_a - k_a.mean(dim=0)
   k_b = k_b - k_b.mean(dim=0)
   cov_k_a = (k_a.T @ k_a) / (N - 1)
   cov_k_b = (k_b.T @ k_b) / (N - 1)
   cov_loss = torch.logsumexp(cut_out_diag(cov_k_a/tau),1).
      mean()/2 +
          torch.logsumexp(cut_out_diag(cov_k_b/tau),1).
             mean()/2

   # loss
   loss = lambda * sim_loss + mu * std_loss + nu * cov_loss

   # optimization step
   loss.backward()
   optimizer.step()
```

**Algorithm 2:** VICReg-ctr PyTorch pseudocode.

```
# f: encoder network,p: projector network, lambda, mu, nu:
   coefficients of the invariance, variance and covariance
   losses, N: batch size, D: dimension of the
   representations, tau: temperature
# mse_loss: Mean square error loss function, relu: ReLU
   activation function, cut_out_diag: remove the diagonal of
    a matrix,

for x in loader: # load a batch with N samples
   # two randomly augmented versions of x
   x_a, x_b = augment(x)

   # compute embeddings
   k_a = p(f(x_a)) # N x D
   k_b = p(f(x_b)) # N x D

   # invariance loss
   sim_loss = mse_loss(k_a, k_b)

   #Make the method contrastive
   k_a = k_a.T
   k_b = k_b.T

   # variance loss
   std_k_a = torch.sqrt(k_a.var(dim=0) + 1e-04)
   std_k_b = torch.sqrt(k_b.var(dim=0) + 1e-04)
   std_loss = torch.mean(relu(1 - std_k_a))/2 + torch.mean(
      relu(1 - std_k_b))/2

   # covariance loss
   k_a = k_a - k_a.mean(dim=0)
   k_b = k_b - k_b.mean(dim=0)
   cov_k_a = (k_a.T @ k_a) / (N - 1)
   cov_k_b = (k_b.T @ k_b) / (N - 1)
   cov_loss = torch.logsumexp(cut_out_diag(cov_k_a/tau),1).
      mean()/2 +
            torch.logsumexp(cut_out_diag(cov_k_b/tau),1).
               mean()/2

   # loss
   loss = lambda * sim_loss + mu * std_loss + nu * cov_loss

   # optimization step
   loss.backward()
   optimizer.step()
```

