# OpenReview forum: "On the duality between contrastive and non-contrastive self-supervised learning"
_ICLR.cc/2023/Conference — ICLR 2023 notable top 5%_

### Official Review · Reviewer_mAZP · 2022-10-14

**Confidence:** 4
**Correctness:** 4
**Technical Novelty And Significance:** 3
**Empirical Novelty And Significance:** 4
**Recommendation:** 8

**Clarity, Quality, Novelty And Reproducibility:**

Clarity:
In general, this paper is well written. Some minor comments/questions:
1. DCL seems undefined when it first appears in Prop 3.1.
2. In Prop 3.1, what's exactly SimCLR and DCL's criteria? It might be good to give a precise definition.
3. In the last paragraph of page 3, what's the meaning of "more than two vectors cannot be the pairwise opposite for SimCLR and DCL's criterion"?

Quality:
The theory is pretty solid and the experiments are extensive.

Novelty:
I think a unifying view of contrastive and non-contrastive methods is pretty novel.

Reproducibility:
I think both the theoretical analyses and the experimental results are reproducible.

**Strength And Weaknesses:**

Strengths:
This is a very insightful paper that provides a unifying view of the contrastive and non-contrastive self-supervised learning methods. From a theoretical perspective, they proved that the sample-contrastive criterion is equivalent to the dimension-contrastive criterion up to the normalization of the embedding matrices. In the experiments, they further empirically verified that the performance of several SOTA sample-contrastive and dimension-contrastive methods can be closed. I believe this unifying view of contrastive and non-contrastive self-supervised learning can help us gain a deeper understanding of these methods. And many techniques and advantages of one family can be transferred to the other, as demonstrated in this paper. It might also be possible to improve the performance by designing a self-supervised learning algorithm that exploits both sample-contrastive and dimension-contrastive criteria.

Weaknesses:
For the theoretical part, it's proved that the sample-contrastive criterion is equivalent to the dimension-contrastive criterion when the embedding matrix is doubly normalized. However, when the embedding matrix is not doubly normalized, the bounds in Lemma 3.4 and Corollary 3.4.1 is pretty loose. So I wonder if it's possible to prove tighter bounds in some more restricted theoretical settings.

**Summary Of The Paper:**

1. This paper interprets the covariance regularization-based non-contrastive methods as dimension-contrastive methods and interprets the traditional contrastive methods as sample-contrastive methods. It then proves that the dimension-contrastive criterion is equivalent to the sample-contrastive criterion when both the rows and columns of the embedding matrix are normalized. It also bounds the difference between these two criteria when the embedding matrix is not doubly normalized.
2. In the experiments, this paper shows that the performance of several SOTA sample-contrastive and dimension-contrastive methods can be closed. This paper showed several design choices have a negligible impact on performance by studying the interpolation between VICReg and SimCLR. Along the way, it also improved the robustness to embedding dimensions of VICReg and improved the performance of SimCLR to match VICReg. It also showed that the normalization strategy does not have a significant influence on performance.

**Summary Of The Review:**

This is a very insightful paper that provides a unifying view of contrastive and non-contrastive methods. It interprets them as sample-contrastive methods and dimension-contrastive methods and proved they are equivalent when the embedding matrix is doubly normalized. In the experiments, they showed that the performance of several SOTA sample-contrastive and dimension-contrastive methods can be closed and many techniques originally developed for one family can be transferred to the other. I believe this unifying view can help us have a deeper understanding of self-supervised learning and guide the community to develop better algorithms.

---

> ### Author Response · Authors · 2022-11-18
> **Response to review**
>
> > However, when the embedding matrix is not doubly normalized, the bounds in Lemma 3.4 and Corollary 3.4.1 is pretty loose. So I wonder if it's possible to prove tighter bounds in some more restricted theoretical settings.
>
> Indeed the bounds are pretty large, even though they are tight. This is due to the lack of assumptions in our results, and we verify empirically in supplementary section G that in practice the values stay very close to the lower bounds.
>
> One assumption that can help derive closer bounds is to consider an embedding matrix with entries i.i.d. From a $ N(0,\sigma^2)$ distribution. The i.i.d. Assumption is rather strong, but the normal distribution is well suited for our problem setting, as it gives a possible good solution to our losses. Without going too much into detail (we would be happy to elaborate further if needed), normalizing such an embedding matrix will lead to a uniform distribution on the hypersphere in the limit case. This is a minimizer of SimCLR's repulsive force and so is a plausible solution. Similarly, since all entries are independent, the embeddings' covariance matrix will also be diagonal (in expected value).
> This means that this simple setting can encapsulate some solutions of the losses that we describe. Now, let's look at what happens to our results when considering this setting.
>
>
> Recall that for a given random variable $X \sim N(0,\sigma^2)$ we have
> $ E[X^2] = \sigma^2$ and $E[X^4] = 3 \sigma^4 $
>
> Using the independence of the X's and the linearity of expected value we obtain
> $$ E[||X||_2^4] = E[X_1^4] +... + E[X_N^4] + E[X_1^2]E[X_2^2]... = 3 N \sigma^4 + N(N-1) \sigma^4 = (2N+N^2) \sigma^4 $$
> This gives us a gap between between our two criteria of $ (NM^2 - MN^2) \sigma^4 $
>
> This also means that the gap is constant, and even 0 if M= N, making the two criterion equivalent from an optimization point of view. However, one limitation is that here we are already considering "good" embedding matrices, and do not consider their dynamics during training, which means that such a result is very limited in its capacity to represent training.
>
> Characterizing realistic embedding matrices during training would be an interesting line of work to design more realistic theoretical analysis of self-supervised learning training dynamics.
>
>
>
>
> **Clarity:**
>
> > DCL seems undefined when it first appears in Prop 3.1.
> In Prop 3.1, what's exactly SimCLR and DCL's criteria? It might be good to give a precise definition.
>
> Thank you for raising this point, we have added a description of all losses considered in supplementary section A.
>
> > In the last paragraph of page 3, what's the meaning of "more than two vectors cannot be the pairwise opposite for SimCLR and DCL's criterion"?
>
> We wanted to convey the fact that for more than two vectors $\mathbb{R}^n$, it is impossible for the dot products of all pairs of vectors to be -1. This mainly gives an intuition as to why the InfoNCE criterion leads to negative pairs with expected dot product of 0 and no -1.

---

### Official Review · Reviewer_Yvpx · 2022-10-24

**Confidence:** 4
**Correctness:** 2
**Technical Novelty And Significance:** 3
**Empirical Novelty And Significance:** 3
**Recommendation:** 5

**Clarity, Quality, Novelty And Reproducibility:**

Clarity:
The paper needs significant clarifications in many places. For example, the focus of the paper is on SimCLR and VICReg, however, there is not even a formal description for these methods. The authors are not supposed everyone ones these methods. Even they do, you also need to describe them so that they match your notation.

Also, as mentioned above, how Definition 3.2 can include all contrastive and non-contrastive learning methods is totally unclear and very confused. This part should also be carefully revised.

Quality and Novelty:
The overall quality of the paper is fair. The paper provides insightful theory and experiments for making contributions towards understanding self-supervised learning. Experiments that substantiate the equivalence of the several invariance criteria, justifying the focus
on only the contrastive criteria may be required. Further experiments towards the nature of embeddings learned and performance without classification heads may strengthen the claims.

Reproducibility:
The authors have provided the complete set of hyperparameters and models/architectures as well as pseudo-codes and learning rate procedures for reproducibility, without the official implementation.

**Strength And Weaknesses:**

Strength:
The paper conducts many experiments towards substantiating the claims. The contribution is novel, to the extent of my knowledge, making a better understanding of the self-supervised learning methods. The work has potential for further theoretical research.

Weaknesses:
1. The presentation is not clear, which needs significant revision. More details are given in the Clarity section.

2. It seems this paper only considers SimCLR and VICReg, while the title tries to cover the whole contrastive and non-contrastive learning methods.

3. Lack of focus on invariance criteria.
Although the claim is made that different similarity criteria are equivalent from an optimization point of view, experimental evidence in support of the claim is missing. I am not convinced that the differences between contrastive and non-contrastive learning methods only rely on the objective in Definition 3.2. I don't think different contrastive learning methods can be simply represented using definition 3.2, e.g., how can you show SimCLR can be written with Definition 3.2? Thus, I think there are many other factors in different contrastive learning methods, an using the designs in this paper cannot cover these.

4. Lack of KNN evaluation.
The paper speaks of the equivalence of the two contrastive regimes and illustrates the equivalence through the gram and covariance matrices, besides empirical evidence with downstream performance. We see that the off diagonal elements in specifically the gram matrix are significantly lower than their diagonal counterparts. But this begs the question of whether the performance of dimension-contrastive methods are at par with sample-contrastive methods for KNN based evaluation of features. KNN performance based experiments will be helpful towards reinforcing the claim of the equivalence of the two regimes.

5. Claim that the small batch size issues with contrastive methods are misleading (Section 5, Clearing up misconceptions).
The paper adequately demonstrates the merits of hyperparameter tuning and use of a better design choice for the projectors may alleviate the problems with small batch sizes, but it is in contrast with the work of [1] where they identify a systematic bias in the updates in SimCLR. This leads us to believe that a better landscape of hyperparameters or projector architectures may be available for large batches as well, reinstantiating the problem with smaller batch sizes in sample contrastive methods, which may require more experiments in those settings. Furthermore, keeping the architecture consistent for fair comparison, we still witness a drop in VICReg performance as the embedding
dimensions decrease (Figures 1 and S4, Table S5).

[1] Yuan, Zhuoning, et al. "Provable stochastic optimization for global contrastive learning: Small batch does not harm performance." International Conference on Machine Learning. PMLR, 2022.

**Summary Of The Paper:**

The paper makes an attempt at bridging the two broadly classified approaches towards self supervised learning, namely contrastive and non-contrastive. Recognizing that the methods can be viewed as sample contrastive and dimension contrastive respectively, the work categorizes popular contributions in the field under these regimes and attempts to consolidate the approaches by drawing
equivalence. Towards this aim, the paper (i) introduces two criteria characteristic of the regimes and demonstrates the
equivalence between them, (ii) attempts to empirically close the performance gaps between popular approaches under the regimes, given the equivalence, (iii) introduces methods that interpolate between the two regimes to isolate components of the loss functions for targeted study, (iv) attempts to validate the theory by improving the performance of SimCLR to match VICReg's and the robustness of VICReg to embedding dimensions like SimCLR.

**Summary Of The Review:**

The paper attempts to answer the fundamental question of the parallels between sample and dimension contrastive approaches towards self-supervised learning. In that regard, the authors conduct many experiments supporting their theoretical motivations and provide a reproducibility statement for verification. However, the presentation of this paper is rather unclear, and the paper lacks some experiments that may serve to strengthen the claims, and may contain some erroneous in statements regarding the sensitivity of these approaches to batch size or feature dimensions.

---

> ### Author Response · Authors · 2022-11-18
> **Response to review 1/2**
>
> >there is not even a formal description for these methods. The authors are not supposed everyone ones these methods. Even they do, you also need to describe them so that they match your notation.
>
> Thank you for raising this very important point to make our work self-contained. We have added supplementary section A, in the appendix (due to space constraints) where we define all of the losses that we consider.
>
> >It seems this paper only considers SimCLR and VICReg, while the title tries to cover the whole contrastive and non-contrastive learning methods.
>
> This choice was made to simplify the title, and we clarify in the abstract directly which methods we consider. We do not only cover VICReg and SimCLR from the theoretical point of view (see proposition 3.2) but we focused on them for our empirical work as they can be seen as representatives of their category of methods in practice.
>
> > Lack of focus on invariance criteria. Although the claim is made that different similarity criteria are equivalent from an optimization point of view, experimental evidence in support of the claim is missing
>
> As we point out in section 3, the two main criteria are cosine similarity and MSE, both being equivalent for normalized embeddings. VICReg and its variants uses MSE, SimCLR cosine similarity, and Barlow Twins the diagonal of the cross-correlation matrix.
>
> The equivalent performance of VICReg-ctr and SimCLR, two methods that only differ by this criterion and the number of negative samples further highlights the similarities between the two approaches.
>
> Similarly, Barlow Twin's approach is very similar to using dot products between positive pairs and leads to the same performance as VICReg (see table 1 from [1]). This and a forced normalization are the only difference between VICReg and Barlow Twins, further reinforcing that both approaches for invariance are equivalent in practice.
>
> In a more general scenario, the only way to minimize those invariance criterion is to have the embeddings making up the positive pair be equal.
>
> [1] Bardes, Adrien, Jean Ponce, and Yann LeCun. "Vicreg: Variance-invariance-covariance regularization for self-supervised learning." arXiv preprint arXiv:2105.04906 (2021).
>
> > I am not convinced that the differences between contrastive and non-contrastive learning methods only rely on the objective in Definition 3.2. I don't think different contrastive learning methods can be simply represented using definition 3.2, e.g., how can you show SimCLR can be written with Definition 3.2? Thus, I think there are many other factors in different contrastive learning methods, an using the designs in this paper cannot cover these.
>
> Indeed SimCLR cannot easily be written in our framework, which is why we introduce proposition 3.1 to give an intuition that goes beyond empirical evidence. We nonetheless propose a small variant of SimCLR through SimCLR-sq that provides identical performance, while being able to fit in our framework. This can be seen by rewriting the InfoNCE criterion as:
>
> $$ L_{\text{SimCLR}} =  \sum_{i=1}^N -\frac{K_{\cdot,i}^T K_{\cdot,i}'}{\tau}  +\text{LogSumExp}\left(\{ \forall j \neq i  |  K_{\cdot,i}^T K_{\cdot,j}/\tau\} \right) $$
>
> Where the LogSumExp can be seen as an approximation of the max operator. We can thus think of this repulsive as doing negative pair mining and considering one negative pair at a time. While this may be less efficient, this will still lead to a diagonal Gram matrix, just like $ L_c $.
>
> We complement this result by looking at the Gram matrices produced by SimCLR in figure S8, finding that it is indeed close to diagonal, with a bit of noise in the off-diagonal terms. This can be partially explained by the lower "efficiency" of this negative pair mining approach.
>
> It is worth noting that SCL, another contrastive method, uses the exact definition of $L_c$, and also provided similar performance as SimCLR at its time of publication.

---

> > ### Author Response · Authors · 2022-11-18
> > **Response to review 2/2**
> >
> > > Lack of KNN evaluation. [...] KNN performance based experiments will be helpful towards reinforcing the claim of the equivalence of the two regimes.
> >
> > We have added supplementary section E in the appendix which reproduces figure 1 with k-NN classifiers. To be as fair as possible, we use 36 combinations of k and temperature for the k-NN classifier and choose the best one for each method. This gives a figure that is almost identical to figure 1, where we see very small gaps in performance between VICReg, VICReg-exp and VICReg-ctr. Similarly as for the linear setting, these gaps could be closed with better hyperparameter tuning. Nonetheless there is still some instabilities in the results, which tend to be noisier than for our linear setting. Improving this evaluation setup would take more time than we had available, but it still gives a strong signal in its current form.
> >
> > The original SimCLR performs significantly worse than other methods, beyond what hyperparameter tuning could do, and our tuned SimCLR significantly outperforms it, matching the performance of the VICReg based methods. This further reinforces the equivalence and is consistent with the correlation between linear and k-nn accuracy observed in the literature.
> >
> > > Further experiments towards the nature of embeddings learned and performance without classification heads may strengthen the claims.
> >
> > To give another argument for the similarity between methods, we have computed CKA between three different runs of each methods, that achieved identical performance. We also added a random network to give an idea of the lower bound that is obtainable. This is available in supplementary section E. CKA[2] is a similarity measure that can be applied between representations and that uses HSIC to estimate the similarities.
> > We can see that for a given method, two runs of it are as correlated as a run of it and one of another method. For example VICReg-1024 is as correlated to VICReg-2048 and SimCLR-2048. This further reinforces the similarities between the learned representations.
> >
> > While this metric is not perfect, this still gives an unsupervised signal of the similarities between representations.
> > [2] Kornblith, Simon, et al. "Similarity of neural network representations revisited." International Conference on Machine Learning. PMLR, 2019.
> >
> > > Claim that the small batch size issues with contrastive methods are misleading (Section 5, Clearing up misconceptions). The paper adequately demonstrates the merits of hyperparameter tuning and use of a better design choice for the projectors may alleviate the problems with small batch sizes, but it is in contrast with the work of [1] where they identify a systematic bias in the updates in SimCLR.
> >
> > We fixed the abstract to not make it sound like one of our contributions
> > The claims from our experiments only affect the embedding dimension related misconception, and we rely on previous results to discuss the batch size dependence of contrastive methods. Indeed the works that we reference do find an issue in SimCLR, similarly to [3], but with small changes this can be significantly alleviated.
> > Per all of these works, we see that this issue is not inherent to all contrastive methods but mostly to SimCLR, and while our work does not encapsulate the complexity of training induced by the choices made for a given criterion, this is an interesting direction of work to derive better criteria.
> >
> > [3]  Yuan, Zhuoning, et al. "Provable stochastic optimization for global contrastive learning: Small batch does not harm performance." International Conference on Machine Learning. PMLR, 2022.
> >
> >
> > >Furthermore, keeping the architecture consistent for fair comparison, we still witness a drop in VICReg performance as the embedding dimensions decrease (Figures 1 and S4, Table S5).
> >
> > There are multiple factors that can play a role in this drop, but that would require further work to study in depth. First, the use of the RBF kernel in SimCLR compared to the quadratic one from VICReg/SCL can help achieve as distinguishable features with a significantly smaller space, which could be one possible explanation. Studying SCL more carefully could help validate or not this hypothesis, at least empirically.
> >
> > This could also be caused by the negative pair mining interpretation of the InfoNCE criterion, meaning that the repulsive force is focused on a smaller number of points at a time and thus does not need as high of a dimension to produce meaningful results. Again, all of this is handwavy and not meant to give a definitive answer.
> >
> > Studying this phenomenon is an interesting line of work, although it is worth noting that you can still achieve peak performance for all methods in settings that are only negligibly more computationally expensive.

---

> > > ### Author Response · Authors · 2022-12-06
> > > **Potential remaining concerns**
> > >
> > > Since the discussion period is ending soon, we wanted to know if our answers properly addressed your concerns ?
> > > If this is not the case, we would be happy to elaborate further about any remaining concerns you may have.

---

### Official Review · Reviewer_87vt · 2022-10-24

**Confidence:** 4
**Correctness:** 4
**Technical Novelty And Significance:** 4
**Empirical Novelty And Significance:** 3
**Recommendation:** 8

**Clarity, Quality, Novelty And Reproducibility:**

This paper provides an overall novel view of self-supervised learning, which may be useful to provide further insights in how the methods used in this field can be linked to each other. The paper is very clear and easy to understand, and I find it very good in quality overall. The authors also provide details to reproduce their experiments.

**Strength And Weaknesses:**

Strengths:

- This paper promotes an interesting link between two variants of self-supervised loss which are often used in practice. Linking these two variants is a very original and interesting line of research, and the fact that the authors demonstrate that the two are very closely related is very useful to provide a better understanding of these techniques.

- The theoretical part of the paper is useful, in that it shows that each of the two variants of self-supervised loss can be used to derive bounds for the other one. This is done in Equations 6 and 7 in the main paper. While the upper bounds in these equations are quite loose (for example, in Equation 6 the upper bound has a difference of the order $N^2 - N$), I believe that it should be argued that the lower bounds are more important. In Equation 6, normalizing the embeddings means we are optimizing $L_c$, and the lower bound tells us that $L_{nc}$ cannot be arbitrarily large either (similarly in Equation 7). I think this is an important connection between the two styles of self-supervised learning.

- The experimental part of the paper is also insightful, in that it supports the proposition of the authors that the two styles of self-supervised learning are closely related, so with sufficient tuning both should achieve similar performance. This is in line with the author's theoretical results. Moreover, in the Appendix the authors also compare the relationship between the losses in practice. These experiments serve to fully support the close connection between the two variants of self-supervised learning.

Weaknesses:
- The authors demonstrate how several well-known self-supervised learning methods fall in their framework of sample contrastive and dimension contrastive methods. To fully complete this comparison, I believe that methods such as MoCo, BYOL and DINO should also be included in this discussion (in Proposition 3.2), since they are also very well-known. Knowing how these methods align in this framework would be useful (mainly because BYOL and DINO also make use of a second teacher modle during training).

- As a minor issue, for the sake of clarity I believe that in their definitions of $L_c$ and $L_{nc}$, the authors should include the alignment term between the representations, even if it is the same across the two. This would make it easier for the reader to recognize that the main difference across methods is the regularization term used in each.

Minor issues:
There is a typo in Section 5.1, third line: “strateoes” -> “strategies”


**Summary Of The Paper:**

This paper links two different views of self-supervised learning, namely contrastive and non-contrastive self-supervised learning. This is done by interpreting the non-contrastive methods as dimension contrastive ones, and regular contrastive methods as sample contrastive. The authors use this insight to demonstrate relationships between the two types of self supervised loss, and demonstrate how one can derive one type of loss from the other.

**Summary Of The Review:**

I believe that this is a solid work, and barring some minor issues I have with the presentation I believe that it should be accepted in the conference. The framework proposed by the authors can be used to unify several different methods used in self-supervision, and this insight can be potentially used to further improve upon existing methods, by easily considering both contrastive and non-contrastive variations.

---

> ### Author Response · Authors · 2022-11-18
> **Response to review**
>
> >  This is done in Equations 6 and 7 in the main paper. While the upper bounds in these equations are quite loose [..]
>
> Indeed the bounds are quite large as they encapsulate all possible matrices, which are mostly poor solutions to our optimization problem. We empirically argue in supplementary section G that in practice we stay extremely close to the lower bound, which as the reviewer pointed out is the more interesting quantity.
>
>
>
> > I believe that methods such as MoCo, BYOL and DINO should also be included in this discussion (in Proposition 3.2), since they are also very well-known.
>
> For BYOL, the main issue is that the repulsive force between samples is not done through the loss function but through the architecture (asymmetry + stop gradient). This makes it significantly harder to include in our theoretical framework while retaining its simplicity.
>
> For MoCo the main issue comes from the use of the momentum encoder that produces the negative samples. This makes it hard to apply our results with it, even though it is conceptually extremely similar to SimCLR, with "better" negative samples that come from the EMA network.
> One way to link it to our framework is to consider the network near convergence where the EMA network will not be updated as drastically and where it could be approximated by the non-EMA network. In this case the method would fall back to something akin to SimCLR with a queue to increase the number of negative samples.
>
> Regarding DINO and other clustering based methods (SwAV, DeepCluster …), the main limitation is again not conceptual but in the details. The softmax at the output of DINO means that the projector can be seen as doing an InfoNCE type loss between the embeddings from the penultimate layer, and the cluster prototypes that make up the last layer of the projector. The reliance on other anti-collapse mechanisms such as using an EMA teacher and a centering for its outputs makes it even harder to include in our work.
>
> We agree that this is indeed a limitation of our work, as we opted for simplicity in order to make our results more intuitive and easily understandable. Studying how these method can be linked to either the sample-contrastive or dimension-contrastive criterion is an interesting line of work.
>
> >the authors should include the alignment term between the representations, even if it is the same across the two.
>
> We have clarified how criteria are the sum of a regularization and alignment term in section 3, and would be happy to further clarify this part if it is not satisfactory in its current form.
>
> > Minor issues: There is a typo in Section 5.1, third line: “strateoes” -> “strategies”
>
> Thank you for pointing it out, this has been fixed in the revised manuscript.

---

> > ### Comment · Reviewer_87vt · 2022-11-19
> > **Thank you for your responses.**
> >
> > I thank the authors for their response to my comments, and their explanation on how the above self-supervised learning techniques relate to this work.
> >
> > I believe that this explanation is improtant to fully understand their relationship with the theoretical framework of the authors. Including them in the paper improves the framework and better positions it with respect to previous work, even if they do not fully align with the framework the authors propose.

---

### Official Review · Reviewer_6jNg · 2022-10-27

**Confidence:** 3
**Clarity, Quality, Novelty And Reproducibility:** see above
**Correctness:** 4
**Technical Novelty And Significance:** 4
**Empirical Novelty And Significance:** 4
**Recommendation:** 10

**Strength And Weaknesses:**

I am a big fan of unification papers. There are several new self-supervised methods appearing every year each claiming SOTA performance, but every method makes its own design choices and so conceptual comparisons can be difficult. This paper shows that many design choices are not essential, and is able to transform SimCLR into VICReg and vice versa while using the same network architecture. The authors also show that simply tuning some of the SimCLR hyper-parameters increases its performance on ImageNet to the SOTA level. I think this is a very welcome contribution and a very strong accept.

All the experiments were run on ImageNet for 100 epochs (only two final runs were made for 1000 epochs). Given that ImageNet optimization is costly, I was wondering if the authors could also demonstrate some of the results e.g. on CIFAR-10 where running 1000 epochs is not a problem. Or is CIFAR-10 "too simple" for the purpose of this study?

**Summary Of The Paper:**

The paper "On the duality between contrastive and non-contrastive self-supervised learning" suggests a unifying perspective on two families of self-supervised learning. One family is the contrastive methods (such as SimCLR) that repel samples from each other through negative pairs. Another family is the covariance-regularizing methods (such as VICReg) that repel samples from each other by enforcing a unit spherical covariance matrix of the embedding. The two competing families have so far been seen as conceptually distinct. This paper argues mathematically and shows empirically that they are approximately equivalent.

**Summary Of The Review:**

Very strong paper suggesting a unifying perspective on various self-supervised learning methods.

---

> ### Author Response · Authors · 2022-11-18
> **Response to review**
>
> We want to thank the reviewer for their positive feedback, and for raising the question of training time.
>
> > Given that ImageNet optimization is costly, I was wondering if the authors could also demonstrate some of the results e.g. on CIFAR-10 where running 1000 epochs is not a problem. Or is CIFAR-10 "too simple" for the purpose of this study?
>
> While CIFAR-10 is interesting in itself, its significant simplicity compared to ImageNet means that there is not much to be learned from doing both. While 1000 epochs leads results much closer to saturation, we would like to point out that due to the small size of the dataset, 1000 epochs of CIFAR-10 are equivalent to ~40 epochs of ImageNet in terms of images seen.
> Nonetheless we can give some preliminary results on CIFAR-10 in 1000 epochs setting (usign a resnet18 encoder).
> We relied on solo-learn (https://github.com/vturrisi/solo-learn) for these experiments and tried to improve their SimCLR baseline.
>
> | Method| VICReg | SimCLR (original) | SimCLR (tuned) |
> |---|---|---|---|
> | Top-1 (in %) | 92.08 | 90.74 | 91.60 |
>
> As we can see, with the limited time that could be allotted to these experiments, we were able to reduce the gap in performance from 1.3 points to less than 0.5 points. While not as strong as our results on ImageNet, this still shows that our experimental insights can be applied to other datasets.

---

### Public Comment · ~Jason_D._Lee1 · 2022-11-06
**Please comment on relation to https://arxiv.org/abs/2008.01064**

I think there should be some discussion of https://arxiv.org/abs/2008.01064 . Our work already established an equivalence between contrastive and non-contrastive and a way to give a unified theoretical analysis, predating this paper by 2 years.

---

> ### Author Response · Authors · 2022-11-18
> **Response to comment**
>
> In our answer, we will assume that you are mainly talking about Section 6 from your work.
> ACE and CCA are indeed closely related to dimension-contrastive methods such as VICReg, where you could instantiate VICReg as the Lagrangian of the optimization problem defined by ACE, with $\eta = \psi$.
> However there is no mention of sample-contrastive methods in this section, except from a mention that from CCA you can derive error bounds that are similar to the ones derived for SCL in the linear evaluation regime.
> Related to these similar bounds, the following sentence is written "non-linear CCA achieves similar guarantees as spectral contrastive loss, **without needing any negative samples.**" (Emphasis ours). This actually reinforces our contributions since we are able to show that covariance based methods actually use negative samples implicitly and can be rewritten in a clear contrastive manner, VICReg and SCL being the exemplar methods that we study. Nonetheless, the similarity between those bounds is already a first step towards understanding the similarities between methods.
>
> Regarding the unification of methods, a link is made between CCA and SimSiam in the following sentence: "The non-linear CCA problem is almost identical to SimSiam, except that we use normalization of representation instead of stop-gradient to prevent representation collapse."
>
> It is unclear how this makes the methods "almost identical", as there are various mechanisms that can avoid collapse and still provide representations with different properties, most notably reconstruction. See for example CAE https://arxiv.org/abs/2202.03026 , which uses an alignment loss coupled with reconstruction and also suffers from poor linear performance, similarly to methods such as MAE https://arxiv.org/abs/2111.06377.
>
> While the similarities could probably be supported empirically, when looking at how they are supported it is unclear how they limit the novelty of our work.
>
>
> Nonetheless, your work is very relevant to ours and we have added a mention to it in the revised manuscript.

---

### Author Response · Authors · 2022-11-18
**General response to reviews**

We thank the reviewers for their comments that have helped improve the paper, strengthening our contributions. We are glad that they found our work insightful in unifying self-supervised methods, both on the theoretical and empirical side.
While we provide individual answers to each review, we wanted to summarize what has been pointed out in the reviews and how the revised version differs from the original submission.

**Summary of the reviews:**
- Lack of introduction of various methods. We have made this clearer by writing all of the methods' losses with our notation.
- Lack of focus on the invariance criterion. We have clarified how criteria are built to make clearer our choice of focus on the regularization aspect of the loss functions.
- Lack of k-nn evaluation to further justify our results empirically. We have added k-nn results that show similar behaviors as the linear setting, and also added an analysis of how the representations correlate between methods to use a non performance-based metric.


**Summary of our revisions:**
- Added supplementary section A to introduce all loss functions that we consider
- Added supplementary section E to provide additional empirical evidence of our results, with k-nn classifiers and using CKA to measure the correlation between representations
- Made clearer how criteria are built with respect to the invariance criterion in section 3

---

### Comment · Program_Chairs · 2023-03-21
**Congratulations on winning the ICLR 2023 Outstanding Paper Honorable Mention!!**

Hello authors,

Big congratulations! For more details about the award process this year, please see https://blog.iclr.cc/2023/03/21/announcing-the-iclr-2023-outstanding-paper-award-recipients/

Best,

Mengdi, on behalf of ICLR 2023 PCs

---

### Decision · Program_Chairs · 2023-01-20

**Decision:**

Accept: notable-top-5%

**Justification For Why Not Higher Score:**

N/A

**Justification For Why Not Lower Score:**

Self-supervised learning is an important topic of research and the proposed work provides an insightful analysis that can related and understand many previous works on the topic.

**Metareview: Summary, Strengths And Weaknesses:**

This paper provides theoretical and empirical analyses to bridge the gap between contrastive and non-contrastive self-supervised learning approaches. Specifically, the authors introduce the concept of dimension-contrastive learning, and show its equivalence to conventional sample-contrastive learning under the assumptions on the normalization of the embeddings. Then, the authors relate popular self-supervised learning methods using the proposed analysis, and empirically verify that such equivalence holds, by using variations of a dimension-contrastive (VICReg), and a sample-contrastive learning method (SimCLR), further improving their performance based on the findings.

The paper was well-received by most reviewers, with an exception of one reviewer who gave this paper the rating of “borderline reject”. The reviewers considered the work as highly novel and insightful as it showed an important link between seemingly unrelated self-supervised learning methods and proposed theoretical analyses as useful as each of the two variants of self-supervised loss can be leveraged to derive bounds for the other. The reviewers also found the empirical analysis as insightful as it supports the main claim that the two variants of self-supervised learning methods are related, and reveals how to further improve them.

However, at the same time, the reviewers were concerned about not including some popular self-supervised learning frameworks into discussion (e.g. MoCo, BYOL, and DINO), somewhat loose upper bound in the proposed theoretical analysis, missing formal descriptions of the method that are explored in the paper (SimCLR and VICReg). However, they were addressed away in the revised version of the paper, and the authors’ responses.

In sum, this is an exceptional research paper that presents an important finding that can further advance our understanding of self-supervised learning, unifying a large number of previous works. I believe that the paper will be of interest to a large audience at ICLR.


**Note From Pc:**

if the above contains the word "oral" or "spotlight" please see: "oral" presentation means -> notable-top-5% and "spotlight" means -> notable-top-25%. As stated in our emails, we are disassociating presentation type from AC recommendations